# The metabolite-controlled ubiquitin conjugase Ubc8 promotes mitochondrial protein import

Saskia Rödl[1], Fabian den Brave[2], Markus Räschle[3], Büsra Kizmaz[1], Svenja Lenhard[1], Carina Groh[1], Hanna Becker[4] , Jannik Zimmermann[5] , Bruce Morgan[5], Elke Richling[4], Thomas Becker[2], Johannes M Herrmann[1] 

**Mitochondria play a key role in cellular energy metabolism. Transitions between glycolytic and respiratory conditions induce considerable adaptations of the cellular proteome. These metabolism-dependent changes are particularly pronounced for the protein composition of mitochondria. Here, we show that the yeast cytosolic ubiquitin conjugase Ubc8 plays a crucial role in the remodeling process when cells transition from respiratory to fermentative conditions. Ubc8 is a conserved and well-studied component of the catabolite control system that is known to regulate the stability of gluconeogenic enzymes. Unexpectedly, we found that Ubc8 also promotes the assembly of the translocase of the outer membrane of mitochondria (TOM) and increases the levels of its cytosol-exposed receptor subunit Tom22. Ubc8 deficiency results in compromised protein import into mitochondria and reduced steady-state levels of mitochondrial proteins. Our observations show that Ubc8, which is controlled by the prevailing metabolic conditions, promotes the switch from glucose synthesis to glucose usage in the cytosol and induces the biogenesis of the mitochondrial TOM machinery to improve mitochondrial protein import during phases of metabolic transition.**

## Introduction

Mitochondria are of central relevance for cellular metabolism. They house the enzymes needed for respiration, the tricarboxylic cycle, the biogenesis of iron-sulfur clusters, and multiple reactions in the synthesis and breakdown of amino acids, fatty acids, membrane lipids, ubiquinone, and other metabolites (Wallace, 2005). Owing to their crucial role in metabolism, the volume and composition of the mitochondrial network is strongly responsive to changing metabolic conditions, much more so than that of most other cellular compartments (Morgenstern et al, 2017; Tsuboi et al, 2020). The shift from glycolytic fermentation to respiration in baker's yeast is a well-

studied metabolic transition, which is accompanied by a major remodeling of the mitochondrial proteome (DeRisi et al, 1997; Di Bartolomeo et al, 2020). This metabolic transition is called the "diauxic shift" and can be observed in glucose-grown yeast cultures. Cells initially produce ethanol by the fermentative breakdown of glucose until glucose depletion induces a shift to respiration. A large fraction of the genes for mitochondrial proteins are induced by the diauxic shift, leading to a more than twofold increase in the total copy number of mitochondrial proteins (Morgenstern et al, 2017). A concerted transcriptional program leads to the induction of respiratory enzymes and other mitochondrial proteins. In parallel, the gene expression of gluconeogenic enzymes is induced to enable glucose replenishment by ethanol usage (Laz et al, 1984; DeRisi et al, 1997; Galdieri et al, 2010; Liu & Barrientos, 2013). Thus, the "diauxic shift" is mainly driven by responses on the transcriptional level.

The reverse transition from respiratory to fermenting conditions is far less understood and more complicated. In addition to changes in protein expression, many proteins, for example, gluconeogenic enzymes or mitochondrial proteins, need to be reduced in their amount or even completely removed. This depletion is accomplished by a process termed catabolite degradation (Chiang & Schekman, 1991). The glucose-induced proteolysis of fructose-1,6-bisphosphatase (Fbp1) in ethanol-grown cultures was extensively studied and allowed the elucidation of the molecular mechanisms of catabolite degradation. A key factor in this remodeling process is the glucose-induced degradation-deficient (GID) complex, which is conserved among eukaryotes and best studied in yeast. The components of this multi-subunit ubiquitin ligase (E3) were initially identified in a genetic screen for mutants deficient in glucose-induced Fbp1 degradation (Hämmerle et al, 1998; Schüle et al, 2000). A recently solved cryo-electron microscopy structure of the GID complex showed that its 20 protein subunits resemble a large organometallic chelator with a central binding site for substrates and the ubiquitin conjugase (E2) Ubc8 (Sherpa et al, 2021). The conserved Ubc8 protein is homologous to other E2 enzymes (Qin et al, 1991) and seems to work exclusively in the

[1]Cell Biology, University of Kaiserslautern, Kaiserslautern, Germany  [2]Institute of Biochemistry and Molecular Biology, Faculty of Medicine, University of Bonn, Bonn, Germany  [3]Molecular Genetics, University of Kaiserslautern, Kaiserslautern, Germany  [4]Food Chemistry, University of Kaiserslautern, Kaiserslautern, Germany  [5]Biochemistry, Center for Human and Molecular Biology (ZHMB), Saarland University, Saarbrücken, Germany

Correspondence: hannes.herrmann@biologie.uni-kl.de

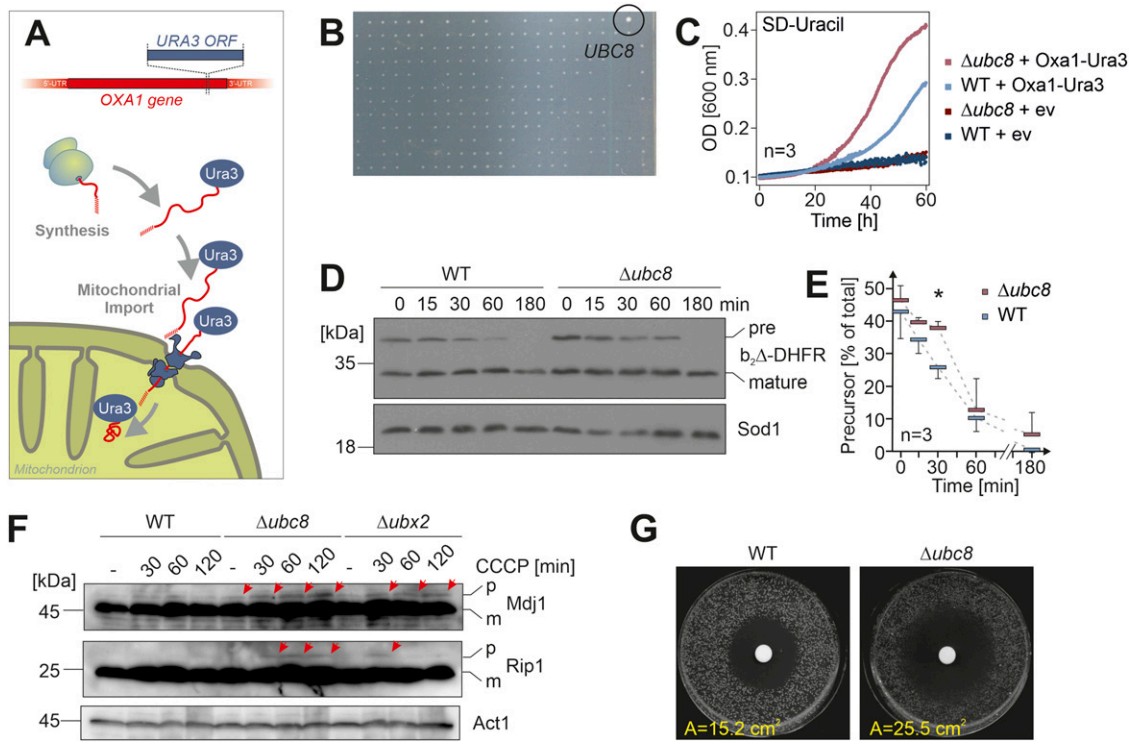

**Figure 1. Loss of Ubc8 leads to the accumulation of mitochondrial precursor proteins in the cytosol.**
**(A)** Schematic representation of the Oxa1-Ura3 screen. The accumulation of the Oxa1-Ura3 precursor in the cytosol confers uracil independence. **(B)** Yeast deletion and DAmP libraries with mutants expressing the Oxa1-Ura3 fusion protein were screened on uracil-deficient plates. Here, a segment of a plate is shown after 1 d of growth at 30°C. The position of the Δ*ubc8* mutant is indicated. See Fig S1B for the entire plate. **(C)** Wild-type (WT) and Δ*ubc8* cells harboring Oxa1-Ura3 expression plasmid or an empty vector for control were grown at 30°C for the times indicated in uracil-deficient synthetic media. Cell densities were measured. The graph shows mean values of three technical replicates. **(D, E)** Wild-type and Δ*ubc8* cells were transformed with a $b_2(1\text{-}167)_{\Delta19}$-DHFR expression plasmid. A galactose-grown preculture was shifted to a galactose-free lactate medium for the times indicated. Aliquots were taken after the times indicated and analyzed by Western blotting using antibodies against DHFR and Sod1 for control. Panel (E) shows mean values and standard deviations of three biological replicates (n = 3). Significance testing was performed with a *t* test (*$P \leq 0.05$). **(F)** Wild-type, Δ*ubc8*, and Δ*ubx2* cells were grown on a galactose medium. Ubx2 removes non-productive import intermediates from the TOM translocase and thus supports efficient protein import into mitochondria (Mårtensson et al, 2019). The uncoupler CCCP was added for the times indicated. Precursor (p) and mature (m) forms of Mdj1 and Rip1 were detected by Western blot. Sod1 served as a loading control. **(G)** Cells were spread on glycerol plates, and a filter was placed in the center to which 10 μl of 10 mM CCCP was added. Cells were incubated for 1 d at 30°C, and the inhibition area (A) was measured. Shown is one representative plate out of several replicates. Source data are available for this figure.

context of GID-mediated protein degradation (Kong et al, 2021). Substrate binding to the constitutively expressed core of the GID complex occurs by substrate-specifying subunits whose expression depends on the prevailing metabolic conditions (Chen et al, 2017; Dong et al, 2018; Kong et al, 2021). Three substrate-specifying subunits have been identified: In the presence of glucose, Gid4 recruits gluconeogenic enzymes such as Fbp1, phosphoenolpyruvate carboxykinase (Pck1), isocitrate lyase (Icl1), and cytosolic malate dehydrogenase (Mdh2). It thereby recognizes specific N-terminal proline motifs in their sequence (Chen et al, 2017). The substrate spectra and recognition motifs of the other specifying factors Gid10 (Melnykov et al, 2019; Langlois et al, 2022) and Gid11 (Kong et al, 2021) are less well understood.

Here, we show that Ubc8 has a second function in addition to its role in catabolite degradation. It promotes the biogenesis of the translocase of the outer membrane of mitochondria, the TOM complex, which serves as the general entry gate for mitochondrial precursor proteins (Shiota et al, 2015; Araiso et al, 2019). In the absence of Ubc8, the central outer membrane receptor Tom22 is diminished, and cells accumulate a partially assembled TOM

complex of compromised function. Our observations identify the E2 protein Ubc8 as an additional regulatory component in the biogenesis of the mitochondrial import machinery, demonstrating that the mitochondrial protein import system is under the cooperative control of enzymes of the phosphorylation and the ubiquitination system.

## Results

### Ubc8-deficient mutants accumulate mitochondrial precursor proteins in the cytosol

Most mitochondrial proteins are synthesized in the cytosol as precursors and subsequently imported into mitochondria (Chacinska et al, 2009). Owing to the very short time between synthesis and import (Williams et al, 2014; Tsuboi et al, 2020), mitochondrial precursors only very transiently encounter the cytosol under physiological conditions. We recently developed a genetic screen to identify yeast mutants with slower import rates

(Hansen et al, 2018). To this end, the orotidine-phosphate decar-boxylase (Ura3) was expressed as a fusion protein with the mito-chondrial protein Oxa1. Cell growth on uracil-deficient plates was only possible upon the cytosolic accumulation of the Oxa1-Ura3 fusion protein, which is promoted under conditions of impaired mitochondrial protein import (Figs 1A and S1A). Using automated mating approaches, the Oxa1-Ura3 expression cassette was in-troduced into yeast libraries covering 4,916 deletion mutants of nonessential genes and 1,102 DAmP (decreased abundance by mRNA perturbation) mutants of essential genes (Schuldiner et al, 2005; Hansen et al, 2018). Eleven of these mutants showed ro-bust growth on uracil-deficient plates and were described before (Hansen et al, 2018). In addition, we observed several extra mutants with a more moderately increased uracil independence, including a strain lacking the ubiquitin conjugase Ubc8 (Figs 1B and S1B and D). Loss of Ubc8 resulted in improved growth in uracil-deficient media indicating higher cytosolic levels of the Oxa1-Ura3 precursor (Figs 1C and S1C).

Here, we asked whether the increased accumulation of the Oxa1-Ura3 precursor observed in Δ*ubc8* cells is indicative of a more general import problem that leads to the cytosolic accumulation of other mitochondrial precursors. To this end, we expressed the well-characterized model protein $b_{2(1-167)\Delta19}$-DHFR in wild-type and Δ*ubc8* cells from a galactose-inducible *GAL* promoter. This fusion protein consists of a matrix-targeting sequence followed by mouse dihydrofolate reductase (DHFR). The fast and rather stable folding of the DHFR domain results in a slow import and transient accu-mulation of the protein in the cytosol (Eilers & Schatz, 1986; Boos et al, 2019). When the galactose-driven expression of the fusion protein was stopped by a switch to a galactose-free medium, the level of $b_{2(1-167)\Delta19}$-DHFR precursor declined over time. However, this decline occurred significantly slower in the Δ*ubc8* mutant than in wild-type cells (Fig 1D and E). This indicates that in the absence of Ubc8, the precursor is either imported more slowly or degraded less efficiently than in wild-type cells. Western blots of whole-cell ex-tracts of the Δ*ubc8* mutant showed indeed precursors of the mi-tochondrial proteins Rip1 and Mdj1; however, their levels were still very low in comparison with those of the mature forms of these proteins (Figs 1F and S2A). We did not observe an induction of the Rpn4-mediated stress response (Boos et al, 2019), which induces genes that are under the control of a proteasome-associated control element. This stress response is characteristic if an import is inhibited by clogger expression (Fig S2B). Nevertheless, Δ*ubc8* cells were hypersensitive to CCCP, which interferes with mito-chondrial import by uncoupling the mitochondrial membrane potential (Fig 1G). However, Ubc8 is not required for growth on respiratory media (Fig S2C) (Qin et al, 1991) or the maintenance of mitochondrial morphology (Fig S2D). In summary, our results in-dicate that the ubiquitin conjugase Ubc8 is directly or indirectly relevant for the efficient depletion of mitochondrial precursor proteins from the cytosol, by promoting either their import or their proteasomal degradation (Fig S1E).

## Ubc8 targets a diverse set of substrates

We next employed mass spectrometry–based proteomics to gain a more comprehensive overview of the proteome dynamics in wild-type and Δ*ubc8* cells. The switch of media in a dynamic stable isotope labeling by amino acids in cell culture (SILAC) approach allows for a precise measurement of the turnover rates of indi-vidual proteins (Ong et al, 2002; de Godoy et al, 2008). This method proved to be very powerful to determine the import, assembly, and degradation of mitochondrial proteins (Bogenhagen & Haley, 2020; Saladi et al, 2020; Schäfer et al, 2022). To this end, we grew wild-type and Δ*ubc8* cells in 2% lactate medium (which promotes respiration and gluconeogenesis) to the mid-log phase with "light" amino acids (i.e., $[^{14}N_2, {}^{12}C_6]$-lysine and $[^{14}N_4, {}^{12}C_6]$-arginine). After removing a first sample ($t_0$), cells were harvested and resuspended in "heavy" (i.e., $[^{15}N_2, {}^{13}C_6]$-lysine and $[^{15}N_4, {}^{13}C_6]$-arginine) medium containing either lactate or lactate plus 2% glucose (Fig 2A). After growth for one doubling time, samples were taken and analyzed by mass spectrometry. Four independent replicates of each sample were analyzed, from which the data were processed and normalized as described in the Materials and Methods section (Table S1). Principal component analysis revealed that the proteome of the Δ*ubc8* and wild-type cells differs considerably even when cells are continuously grown in lactate, which suggests that Ubc8 is of relevance for respiring cells (Fig 2B). The addition of glucose caused a further strong "catabolite effect" on the proteomes of these cells.

The glucose-induced shift from respiration to fermentation in-duces the depletion of gluconeogenic enzymes (Fbp1, Mdh2, Icl1, and Pck1) in wild-type but not in Δ*ubc8* cells (Fig 2C), which is consistent with previous reports (Hämmerle et al, 1998; Schüle et al, 2000). In contrast, if cells were continuously grown in lactate, these enzymes did not differ considerably between wild-type and Δ*ubc8* cells (Fig S3A).

Interestingly, we further noticed that the levels of many mito-chondrial proteins were unexpectedly decreased in Δ*ubc8* cells. This was particularly obvious after hierarchical clustering of the proteomics data, which distinguished different groups of proteins according to differences in their abundance in lactate and glucose media in wild-type and Δ*ubc8* cells, respectively (Fig 2D and E and Table S2). As expected, a defined cluster of gluconeogenic enzymes showed a Ubc8-dependent decline upon glucose addition (Fig 2E, group 4a). This was particularly apparent for the "light" peptides and thus for old proteins, consistent with the regulation by pro-teolysis (Fig 2F). In addition, a clustered group of other glucose-repressed proteins (Fig 2E, group 4b) was defined by the fact that their levels were diminished in Δ*ubc8* cells in comparison with those in wild-type cells even upon continuous growth in lactate (Fig S3B). Interestingly, this group is characterized by an enrichment of mitochondrial (and peroxisomal) proteins, indicating that the levels of many proteins of the IMS, the inner membrane, and the matrix depend on the presence of Ubc8 (Figs 2G and S3C and Table S2). Thus, Ubc8 is not simply a glucose-induced removal factor of four gluconeogenic enzymes but rather plays a much more general role in the adaptation of the cellular proteome to the prevailing met-abolic state.

Ubc8 apparently has a second, distinct role as a factor that promotes the biogenesis of mitochondrial proteins. Interestingly, this latter function was not only apparent upon glucose supple-mentation to respiring cells but also observed when cells were continuously grown in lactate.

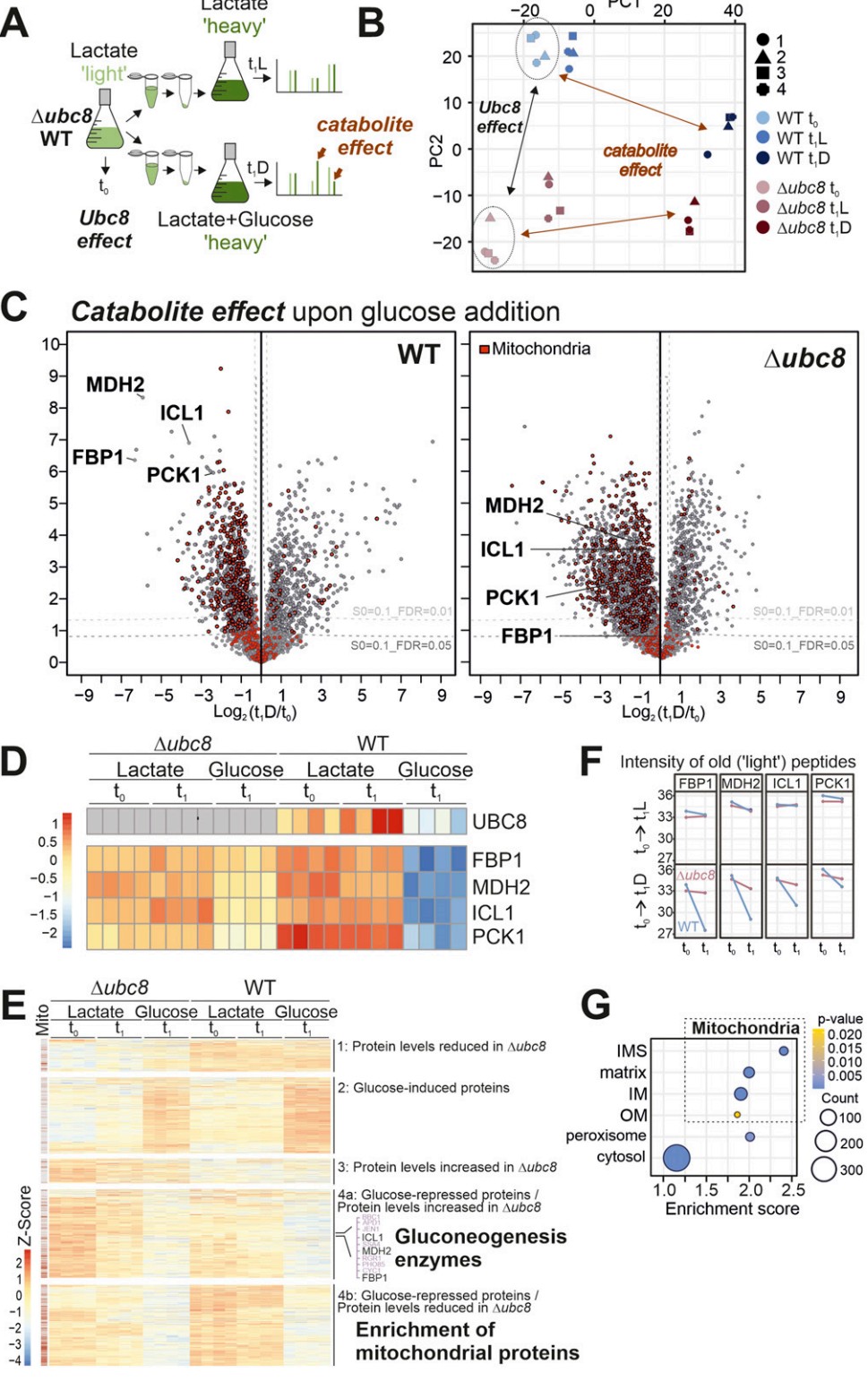

**Figure 2. Ubc8 promotes catabolite degradation and influences levels of mitochondrial proteins.**
**(A)** Schematics of the proteomics workflow. **(B)** Principal component analysis of the data after normalization (see the Materials and Methods section). The entire dataset of the measurement is shown in Table S1. **(C)** Volcano plot comparing the whole-cell proteomes after and before the metabolic shift from lactate to glucose medium of WT and Δ*ubc8* cells. Positions of gluconeogenic enzymes regulated by Ubc8-dependent metabolic degradation are labeled in bold. Mitochondrial proteins are highlighted in red (Morgenstern et al, 2017). Corresponding plots for the samples that were further grown on lactate are shown in Fig S3A. The sum of the "heavy" and "light" intensities was used. **(D, E)** Hierarchical clustering of protein intensities identifies five distinct groups. Positions of mitochondrial proteins are indicated on the left. Gluconeogenic enzymes are found in group 4a. In group 4b, mitochondrial proteins are enriched. Panel (D) selectively shows the heatmaps for gluconeogenic enzymes. See Table S2 for additional information. **(F)** Intensities in the light channel ("old" peptides) of the indicated proteins at $t_0$ and $t_1$ were plotted. Note that lactate-to-glucose switches induce Ubc8-dependent degradation of gluconeogenic enzymes. **(G)** Enrichment scores for proteins in group 4b indicate the presence of many components of mitochondrion-specific GO categories.

## Ubc8 is critical for metabolic remodeling of yeast cells

Ubc8 was initially discovered as a protein required for the rapid glucose-induced degradation of Fbp1 (Schüle et al, 2000).

Consistent with these original reports, we observed that a shift from glycerol to glucose medium induces the rapid depletion of HA-tagged Fbp1 in an Ubc8-dependent manner (Fig 3A). Such a Ubc8-mediated catabolite degradation was also observed for Icl1 and

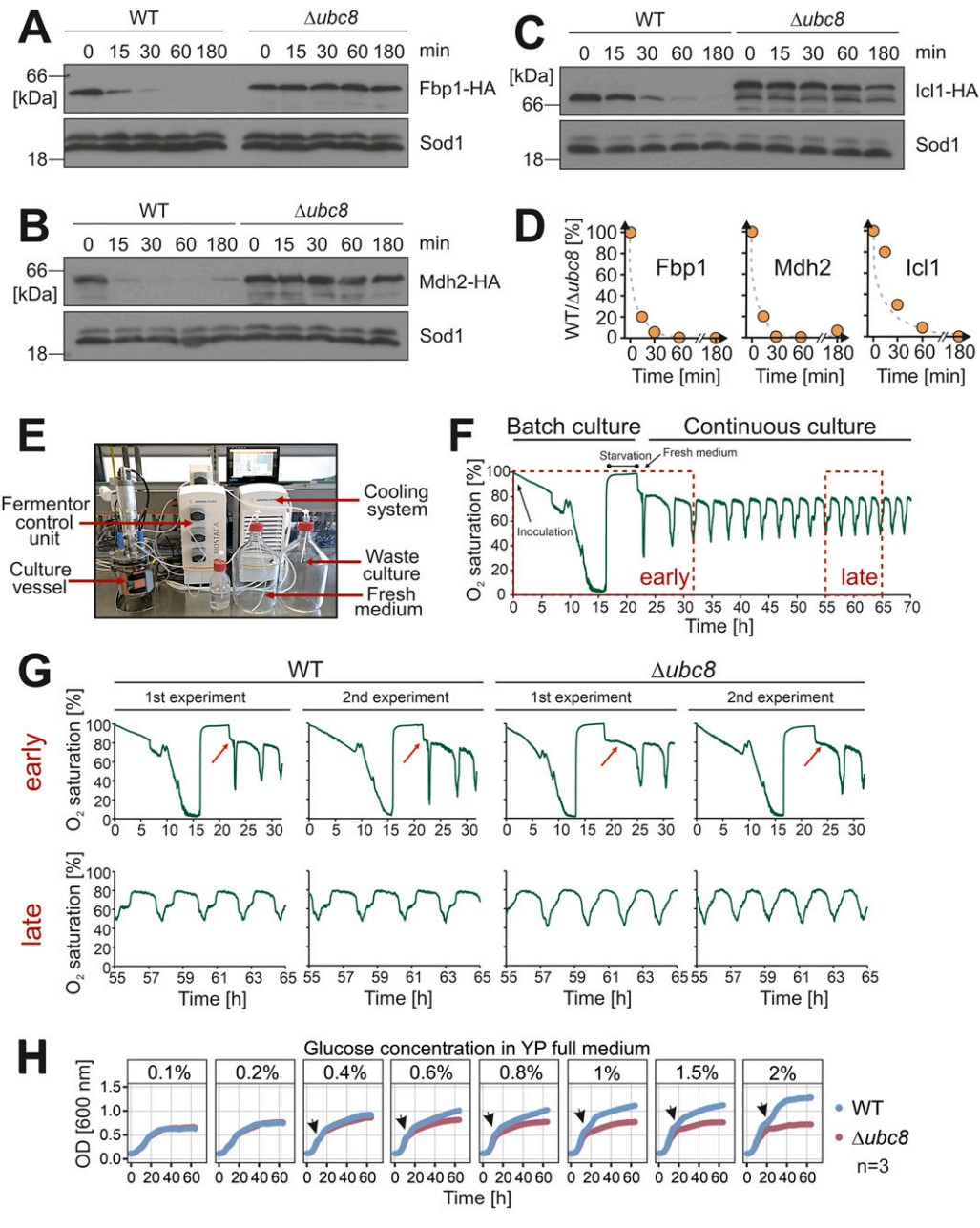

**Figure 3. Ubc8 is crucial for the quick adaptation to metabolic changes.**
**(A, B, C, D)** Wild-type and Δ*ubc8* cells expressing the indicated HA-tagged proteins under endogenous control were cultured in a lactate medium. The medium was replaced by a glucose medium. Aliquots were taken after the times indicated and analyzed by Western blotting. Sod1 served as a loading control. Panel (D) shows the quantification of (A, B, C). **(E)** Setup of the fermentor system. **(F, G)** The fermentor run was initiated by the inoculation of a YPD-grown preculture. Oxygen saturation in the medium was automatically monitored every 10 s. Cells grew as a batch culture until the stationary phase. The culture was then kept in the starvation phase for 5 h to synchronize all cells. Finally, fresh glucose medium was continuously added at defined rates to initiate the metabolic cycling of cultures. The red arrows point to the phase of adaptation between starvation and induction of oxygen consumption that initiates metabolic cycling. Early and late phases of the runs are shown for wild-type and Δ*ubc8* cultures. **(H)** Cultures were grown to the mid-log phase in a glycerol medium and diluted in a medium containing different concentrations of glucose as indicated. Cell densities were continuously measured during 3 d of growth under continuous shaking. Biphasic growth curves are characteristic of diauxic shifts from glucose to ethanol consumption; positions of these diauxic shifts are indicated by arrows.
Source data are available for this figure.

Mdh2 (Fig 3B–D), thereby confirming previous studies (Chen et al, 2017; Karayel et al, 2020; Chen et al, 2021).

Even though the role of Ubc8 in the catabolite degradation of Fbp1 and other gluconeogenic enzymes is well documented, little is known about the physiological relevance of Ubc8 (and the GID complex). Because yeast cells are typically cultured under static growth conditions, the adaptation to varying metabolic conditions is normally of little relevance. However, under defined growth

conditions in a continuous fermentor-based culture, yeast cell populations spontaneously and stably synchronize in a phenomenon known as the yeast metabolic cycle (YMC) (Tu et al, 2005; Slavov et al, 2011; Amponsah et al, 2021). The mechanistic basis for this synchronization remains unclear. The most distinctive feature of the YMC is a stable, population-synchronized oscillation between low- and high-oxygen consumption phases, termed LOC and HOC, respectively. This can be readily monitored by the impact on culture oxygen level, which is inversely proportional to the oxygen consumption rate. The cell division cycle is also synchronized during the YMC (Chen et al, 2007; Slavov et al, 2011; Amponsah et al, 2021). During LOC, cells are not dividing and tend to accumulate storage carbohydrates such as trehalose. On the contrary, during HOC, cells liberate sugar stores and divide (Burnetti et al, 2016; O'Neill et al, 2020). Consistent with these observations, proteomic analyses of YMC-synchronized populations recently showed cyclical changes in the levels of gluconeogenic enzymes, including Mdh2 and Fbp1 during the YMC, with their highest levels during LOC, consistent with the generation of storage carbohydrates (O'Neill et al, 2020). We thus considered that the YMC would be an excellent model to study the impact of Ubc8 deletion in a context where there is regular and periodic switching between gluconeogenesis and glycolysis-dominated metabolism. After initiation of a fermentor run by the injection of a glucose-grown preculture, yeast cells grow as a batch culture and deplete glucose and oxygen before ultimately entering a starvation phase (Fig 3E and F). When a fresh glucose-containing medium is pumped into the vessel after several hours of starvation, the oscillation of the oxygen saturation in the culture medium is spontaneously induced. Importantly, oxygen saturation in the medium is a direct consequence of a population-synchronized oscillation in oxygen consumption (Tu et al, 2005). Periodic changes in oxygen consumption in continuous and synchronized yeast cultures are the most obvious and distinctive feature of the YMC. To test for the relevance of Ubc8 in this context, we deleted the *UBC8* gene in the CEN.PK113-1A yeast strain (Burnetti et al, 2016; Amponsah et al, 2021) and monitored oxygen consumption in a fermentor for several days. Therefore, we noticed that although both wild-type and Δ*ubc8* cultures exhibited the characteristic oxygen saturation cycling, the cycles in the Δ*ubc8* strain only started after a 4-h delay, which was not observed in wild-type cells (Fig 3G, red arrows). Furthermore, the cycles in Δ*ubc8* cells tend to be shorter (Fig 3G, lower panels), which is consistent with a disrupted ability to switch between different metabolic states. Still, Δ*ubc8* cells are able to alternate rhythmically between oxygen-consuming and fermenting phases.

Next, we tested the growth of cells after switching from a glycerol medium to different concentrations of glucose (Fig 3H). We observed no difference in the growth of wild-type and Δ*ubc8* cells when the glucose concentrations were low. This was surprising because we expected that the presence of futile cycles, owing to the simultaneous presence and activity of glycolytic and gluconeogenic enzymes in Δ*ubc8* cells, would waste energy and should negatively affect cell growth, particularly when carbon sources are scarce. However, at higher glucose concentrations, wild-type cells grew to much higher cell densities than Δ*ubc8* cells did. This was not caused by the production of toxic byproducts such as methylglyoxal (Fig S4A and B). Methylglyoxal is predominantly produced by the

dephosphorylation of intermediates during glycolysis. Because methylglyoxal is highly reactive, it forms adducts with lipids, DNA, and proteins and thereby causes the formation of advanced glycation end products (AGEs), which are toxic. Because futile cycles of glycolysis and gluconeogenesis might increase the production of this toxic metabolite, we measured the methylglyoxal concentrations in wild-type and Δ*ubc8* cells 1 and 24 h after switching the cells from 0.2 to 2% glucose (Kalapos, 1999). For the Δ*glo1* mutant, which served as a positive control, increased methylglyoxal levels could only be measured for the 1-h but not for the 24-h time point. This was expected because Glo1 is indeed involved in one of the main detoxification pathways for methylglyoxal, but the toxic metabolite can also be detoxified in Glo1-independent pathways (Murata et al, 1985; Aguilera & Prieto, 2001).

Interestingly, we noticed that the growth curves of wild-type and Δ*ubc8* cells were identical for the first few hours, but the Δ*ubc8* cells lagged behind after the diauxic shift, i.e., the switch from fermentative ethanol production to ethanol-consuming respiration. The diauxic shift is accompanied by dramatic rearrangements of mitochondrial function and structure in yeast cells (Di Bartolomeo et al, 2020). Furthermore, we noticed increased lethality of Δ*ubc8* cells in stationary, glucose-grown cultures (Fig S4C), consistent with problems under metabolic conditions that rely on mitochondrial activity (Ocampo et al, 2012). Thus, these physiological data indicate that the deletion of *UBC8* shows consequences that point to problems in mitochondrial functionality.

### Efficient TOM assembly depends on Ubc8

Because of the reduced functionality of mitochondria in Δ*ubc8* cells and the increased amounts of mitochondrial precursor proteins in the cytosol, we directly tested whether Ubc8 is relevant for protein import into mitochondria. To this end, we isolated mitochondria from wild-type and Δ*ubc8* cells and performed in vitro import reactions with radiolabeled mitochondrial precursor proteins. For this, we used the inner membrane protein Oxa1 (which uses the TOM-TIM23 import pathway) and the ATP/ADP carrier Pet9 (which embarks on the TOM-TIM22 import route). Both proteins were imported with considerably reduced efficiency into Δ*ubc8* mitochondria (Fig 4A), indicating that Ubc8 is important for the biogenesis or stability of the mitochondrial protein import system.

To identify the reason for the import defect of Δ*ubc8* cells, we shifted wild-type and Δ*ubc8* cells from glycerol to glucose medium for 4 h and analyzed their proteomes by mass spectrometry (Figs 4B and S5A and B and Table S3). After the metabolic shift, but not before, Δ*ubc8* cells showed considerably increased levels of gluconeogenic enzymes, as expected. In addition, we noticed that the absence of Ubc8 caused diminished levels of the mitochondrial protein Tom22, which was less pronounced in glycerol-grown cells.

Western blot experiments of isolated mitochondria and whole-cell lysates confirmed the reduced Tom22 levels (Figs 4C and S5C and D), whereas the levels of the pore-forming subunit of the TOM complex, Tom40, and those of the receptor Tom70, remained unaffected. Mitochondria isolated from Δ*gid4* and Δ*gid10* cells also showed reduced Tom22 levels (Fig.4D). Gid4 and Gid10 serve as substrate-binding subunits of the GID complex (Chen et al, 2017; Melnykov et al, 2019; Sherpa et al, 2021). To exclude that the reduced

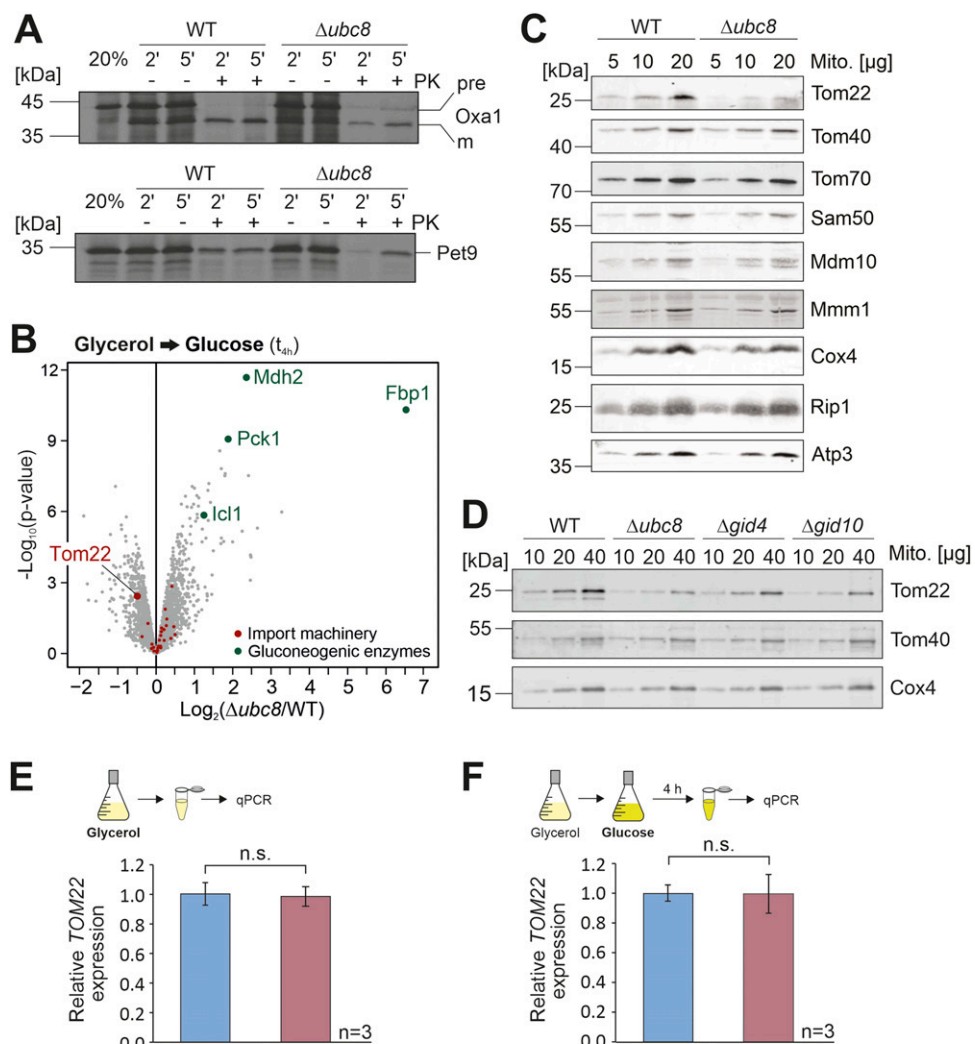

**Figure 4. Absence of Ubc8 or other components of the GID complex leads to diminished levels of the outer membrane protein Tom22.**
**(A)** Radiolabeled Oxa1 and Pet9 were incubated with mitochondria isolated from wild-type or Δ*ubc8* cells grown in galactose medium at 30°C for different times. Non-imported proteins were degraded by the addition of proteinase K (PK). 20% of the radiolabeled proteins used per import reaction are loaded for control. Proteins were visualized by autoradiography. **(B)** Cells were grown in a glycerol medium before glucose was added for 4 h. Cells were subjected to mass spectrometry, and the levels of different groups of mitochondrial proteins were analyzed. Four replicates were analyzed. See also Fig S5A and B and Table S3. **(C, D)** Cells were grown in a glycerol medium. After the addition of glucose for 4 h, mitochondria were isolated and subjected to SDS–PAGE. The indicated proteins were visualized by Western blotting. **(E, F)** *TOM22* mRNA levels of cells continuously grown in glycerol (E) or after the addition of glucose for 4 h (F) were analyzed by qRT-PCR. Shown is the *TOM22* expression in Δ*ubc8* cells relative to the expression in wild-type cells. Mean values and standard deviations of three biological replicates (n = 3) were calculated. Significance testing was performed with a *t* test (n.s., not significant).
Source data are available for this figure.

Tom22 protein levels in Δ*ubc8* cells are the result of transcriptional changes, we measured *TOM22* mRNA levels by qRT-PCR. To this end, either wild-type and Δ*ubc8* cells were continuously grown in a glycerol medium (Fig 4E) or 2% glucose was added for 4 h before samples were taken (Fig 4F). Under both conditions, the deletion of *UBC8* did not affect the expression levels of *TOM22*.

The overexpression of Ubc8 also reduced the levels of Tom22 and the import competence of mitochondria and resulted in an impaired growth on non-fermentable carbon sources (Fig S6A–E). Apparently, the activity of the GID complex is not simply regulated by the levels of the Ubc8 ubiquitin conjugase, consistent with previous findings about the rate-limiting role of the different substrate-binding GID subunits Gid4, Gid10, and Gid11 (Chen et al, 2017; Melnykov et al, 2019; Kong et al, 2021; Langlois et al, 2022). Their specific relevance for the biogenesis of Tom22 will have to be studied in the future.

In summary, our observations show that the GID complex, for which Ubc8 serves as the ubiquitin conjugase, influences the levels of the crucial TOM protein Tom22 in mitochondria, which is a rate-limiting factor for the mitochondrial protein import system (van

Wilpe et al, 1999; Schmidt et al, 2011; Shiota et al, 2011; Zeng et al, 2019).

## Ubc8 facilitates the assembly of Tom22 into functional TOM complexes

Next, we analyzed the size of the TOM complex by blue native gel electrophoresis followed by Western blotting (Fig 5A). We prepared mitochondria from cells that were shifted from glycerol (respiration) to glucose (fermentation) medium for 4 h. In wild-type mitochondria, the TOM complex migrates like the 440 kDa molecular weight marker. In contrast, in Δ*ubc8* mitochondria, Tom40 and Tom22 were part of a faster migrating complex.

Again, we observed considerably reduced levels of Tom22. Given the rather small size shift, the modified complex in Δ*ubc8* mitochondria might lack individual Tom22 subunits, potentially also some of the small Tom proteins (Shiota et al, 2015; Araiso et al, 2019). The sizes of other complexes containing other mitochondrial proteins (cytochrome oxidase and SAM complex) were not altered. However, we also noticed lower levels of the SAM-Mdm10 complex

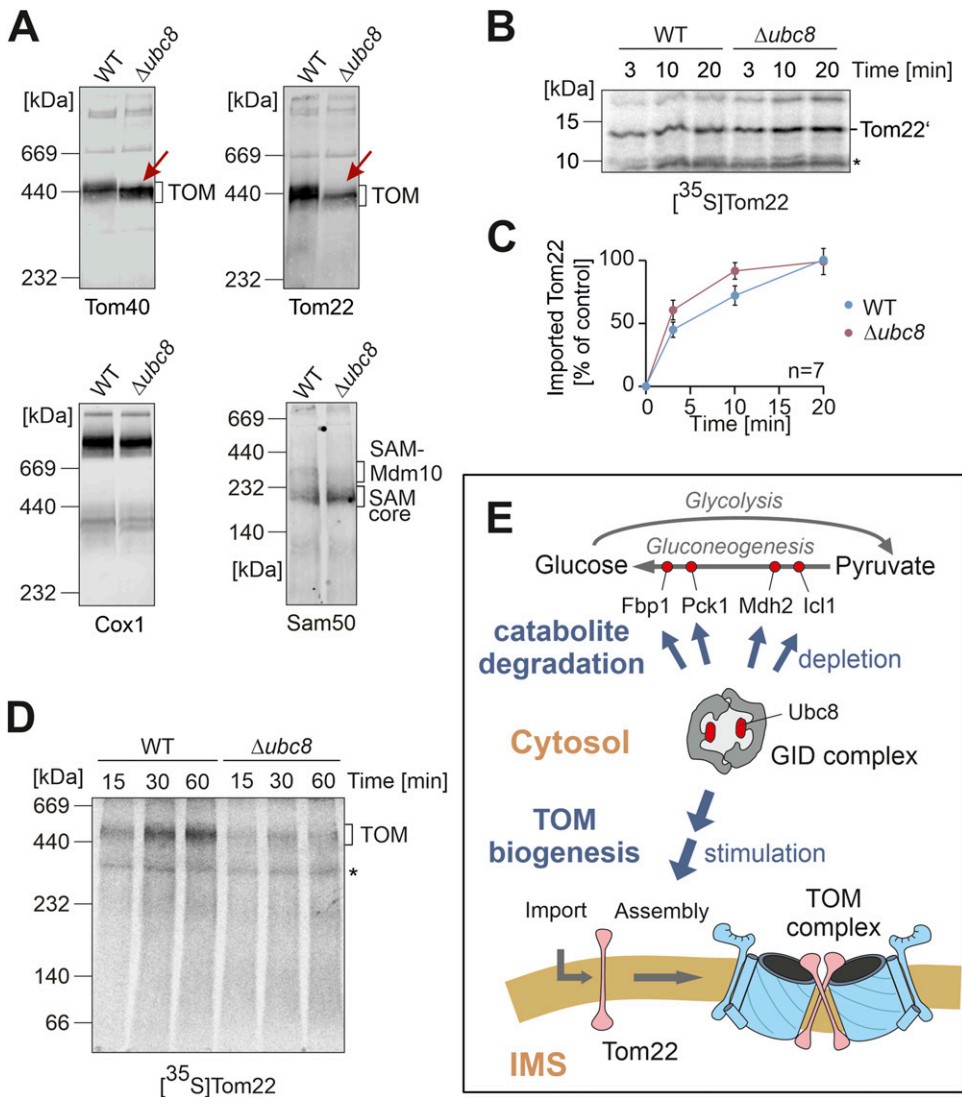

**Figure 5.  Ubc8 is required for the full assembly of the TOM complex.**
**(A)** Cells were grown in a glycerol medium. After the addition of glucose for 4 h, the mitochondria were isolated and subjected to blue native gel electrophoresis. The indicated proteins were visualized by Western blotting. **(B, C)** Radiolabeled Tom22 was imported into wild-type and Δ*ubc8* mitochondria for the indicated time periods followed by incubation with proteinase K to remove non-imported precursor proteins. The membrane-inserted, protease-protected fragment of Tom22 (Tom22′) was analyzed by SDS–PAGE and autoradiography. Unspecific cross-reactions of the antibody are indicated with an asterisk (*). For quantification, the signal of imported Tom22 of the longest time period into wild-type mitochondria was set to 100% (control). Quantification shows mean values and standard deviations of seven independent experiments (n = 7). **(D)** Radiolabeled Tom22 was incubated with mitochondria isolated from wild-type and Δ*ubc8* cells for the times indicated. Assembly of Tom22 into the TOM complex was analyzed by blue native gel electrophoresis and autoradiography. The position of the TOM complex is indicated. Note that the assembly of Tom22 into Δ*ubc8* mitochondria occurs with reduced efficiency. Unspecific cross-reactions of the antibody are indicated with an asterisk (*). **(E)** Model of Ubc8 function. See text for details. Source data are available for this figure.

in Δ*ubc8* mitochondria. The SAM-Mdm10 complex promotes the assembly of Tom22 with Tom40 and the small Tom proteins into the mature TOM complex (Yamano et al, 2010; Ellenrieder et al, 2016).

The assembly of Tom22 with other subunits of the TOM complex is well characterized. The efficiency of Tom22 assembly depends on a complex regulatory network of cytosolic kinases and phosphatases (Schmidt et al, 2011; Rao et al, 2012; Gerbeth et al, 2013; Kravic et al, 2018) and its interaction with several integral factors of the mitochondrial outer membrane (Keil & Pfanner, 1993; Meisinger et al, 2004; Becker et al, 2011; Qiu et al, 2013; Ellenrieder et al, 2016; Wu et al, 2018; Sakaue et al, 2019). To test which of these steps are Ubc8-dependent, we first assessed the translocation step of Tom22 into the outer membrane. To this end, we used a radiolabeled Tom22 variant that was C-terminally extended by three additional methionine residues to improve the detection of a characteristic protease-resistant Tom22′ fragment (Ellenrieder et al, 2016). Wild-type and Δ*ubc8* cells were shifted from glycerol to glucose medium for 4 h before mitochondria were isolated. These mitochondria

were incubated with the radiolabeled Tom22 variant before mitochondria were treated with protease (Fig 5B and C). In wild-type and Δ*ubc8* mitochondria, the characteristic Tom22′ fragment accumulated with the same efficiency. Thus, Ubc8 was not relevant for the insertion of Tom22 into the outer membrane.

Next, we assessed the assembly step of Tom22 into the TOM complex, again using in vitro import experiments with mitochondria from glycerol-to-glucose-switched wild-type and Δ*ubc8* cells, and monitored the assembly of Tom22 into the TOM complex by blue native gel electrophoresis (Fig 5D). Therefore, we noticed that Tom22 assembly was decreased in the absence of Ubc8, indicating that the reduced Tom22 levels were because of impaired assembly of Tom22 into the TOM complex. The reduced levels of the SAM-Mdm10 complex in Δ*ubc8* mitochondria (Fig 5A) could contribute to the impaired Tom22 biogenesis.

In conclusion, the presence of Ubc8 is obviously important to facilitate the biogenesis of the TOM complex, particularly after metabolism switches from respiration to fermentation. If Ubc8 is

absent, only a partially assembled and functionally compromised TOM complex is formed, which leads to a moderate depletion of mitochondrial proteins, in particularly those of the matrix for which the Tom22 levels seem particularly important (Fig S5E). Thus, the biogenesis of the major protein entry site of the mitochondria, the TOM complex, is under the control of the ubiquitin conjugase Ubc8 (Fig 5E). To our knowledge, this is the first observation that the cytosolic ubiquitin system promotes the biogenesis of the mitochondrial import machinery.

## Discussion

Cells dynamically shape their proteome in response to the prevailing growth conditions. In particular, metabolic changes can induce a massive reorganization of the cellular proteome (Nielsen, 2017). For example, in yeast cells, the addition of glucose to galactose-grown cells changes the expression of 25% of all gene products by more than twofold, even though neither growth rates nor cell sizes are affected (Ronen & Botstein, 2006). In a nutshell, proteome remodeling is the result of altered gene expression on the one hand and regulated protein degradation on the other. Thus, the orchestrated interplay between transcription factors and components of the ubiquitin-proteasome system is crucial for dynamically and accurately adapting the proteome.

We designed a quantitative proteomics experiment based on the dynamic SILAC (Pratt et al, 2002) to follow the changes in the yeast proteome induced by switches from respiratory glucose synthesis to fermentative glucose consumption. The dynamic SILAC has been successfully used in the past to measure the rates of protein turnover on the basis of the relative loss of "old" (in our case, light) peptides (Doherty et al, 2009; Dorrbaum et al, 2018; Mathieson et al, 2018; Saladi et al, 2020) or the rates of protein synthesis based on the accumulation of "new" (in our case, heavy) peptides (Schäfer et al, 2022). Our results document the potential of metabolic labeling for both aspects:

1. On the one hand, we observed a rapid Ubc8-dependent depletion of gluconeogenic enzymes upon glucose addition, resulting in the rapid loss of "old" peptides of Fbp1, Mdh2, and Icl1 in wild-type but not in Δ*ubc8* cells. This is consistent with the established role of Ubc8 in the catabolic degradation of gluconeogenic enzymes (Schüle et al, 2000; Chen et al, 2017, 2021; Karayel et al, 2020; Sherpa et al, 2021). Even though Δ*ubc8* cells had considerably higher levels of gluconeogenic enzymes, their growth rates remained unaffected even when glucose levels were low. Thus, futile cycles, caused by the simultaneous presence of glycolytic and gluconeogenic enzymes, might either be also prevented by other means such as posttranslational modifications and allosteric regulation of the enzymes (Dihazi et al, 2003; Smets et al, 2010), or do not directly impair growth.

2. On the other hand, we unexpectedly detected that the absence of Ubc8 strongly impairs the biosynthesis of mitochondrial proteins, proposing a role of Ubc8 in mitochondrial biogenesis. The dynamic SILAC method made it possible to discriminate between changes in protein expression and those in protein stability within the same experiment. In this study, we confirmed the role

of Ubc8 for the general accumulation of mitochondrial proteins by a second proteomics experiment using a simpler label-free quantification procedure (see Figs 4B and S7) in which the quantification was more straightforward. However, this second experiment did not allow us to tell whether mitochondrial proteins were less produced or faster degraded.

The absence of Ubc8 considerably reduced the levels of the mitochondrial outer membrane protein Tom22. The biogenesis of Tom22 is controlled both on the level of its import into the outer membrane and on the level of its assembly into the TOM complex. Previous studies by Chris Meisinger and coworkers described a glucose-dependent regulation of the import and assembly of Tom22 in depth (Schmidt et al, 2011; Gerbeth et al, 2013; Kravic et al, 2018). In the cytosol, Tom22 is phosphorylated by casein kinases CK1 and CK2 and by the PKA. The most abundant outer membrane protein porin (Por1) modulates the assembly of Tom22 into the TOM complex (Ellenrieder et al, 2016; Sakaue et al, 2019). Therefore, Por1 was proposed to slow down the incorporation of Tom22. Potentially, Ubc8 accelerates proteolytic degradation of this Por1 species. Interestingly, Por1 was indeed identified as a potential substrate of the substrate adaptor Gid10 that specifies proteins for GID-mediated degradation (Langlois et al, 2022). However, Ubc8 could also modulate the SAM-Mdm10–mediated biogenesis of Tom22. It will be interesting to test how Ubc8-mediated degradation directly interferes with this intricate assembly of the TOM complex.

Apparently, Tom22 represents the critical key element in the regulation of mitochondrial protein import. What makes Tom22 so special? Tom22 is a central component of the TOM complex that serves as "multifunctional organizer" (van Wilpe et al, 1999). This term appropriately describes the function of Tom22, as in the absence of Tom22, the TOM complex dissociates in core units that mainly consist of the β-barrel protein Tom40. The addition of Tom22 converts these pores into fully functional dimeric and trimeric TOM complexes, thereby considerably enhancing protein translocation efficiency (van Wilpe et al, 1999; Shiota et al, 2015; Bausewein et al, 2017; Araiso et al, 2019; Tucker & Park, 2019; Zeng et al, 2019; Wang et al, 2020). Recent studies indicated that the Tom22 association drives a dynamic conversion of different functional states of the TOM complex, which modulates its substrate preference (Gornicka et al, 2014; Sakaue et al, 2019). Thus, the controlled incorporation of Tom22 into the TOM complex seems to be an excellent strategy to modulate mitochondrial import capacity on a general scale (Araiso et al, 2021). Interestingly, although Ubc8 induces Fbp1 degradation exclusively upon glucose addition, Ubc8 improved Tom22 assembly also under continuous non-fermentative conditions, albeit to a much smaller degree (Fig S7 and Table S4). This suggests that the Ubc8 effect on mitochondrial protein import is not exclusively mediated via the glucose-controlled substrate-specifying factor Gid4. It appears likely that Ubc8 and the GID complex control the assembly of the TOM complex also via other substrate-specifying factors, potentially via Gid10. It will be exciting in the future to unravel the molecular details by which the phosphorylation and the ubiquitination systems in the cytosol cooperate to control mitochondrial protein biogenesis.

# Materials and Methods

## Yeast strains and plasmids

All yeast strains used in this study are based on the wild-type strains BY4742 (MATα *his3 leu2 lys2 ura3*), YPH499 (MATa *ura3 lys2 ade2 trp1 his3 leu2*), YPH499 Δ*arg4* (MATa *ura3 lys2 ade2 trp1 his3 leu2 arg4*) (Woellhaf et al, 2016), or CEN.PK113-1A (MATα) (Sikorski & Hieter, 1989). To delete *UBC8* and *GID10*, the genes were chromosomally replaced by kanMX4 or natNT2 cassettes using pFA6α-kanMX4 or pFA6α-natNT2 as templates. For HA-tagging of Fbp1, Mdh2, and Icl1, the sequence of 6HA-natNT2 was amplified from the plasmid pYM17 and genomically integrated downstream of the respective genes (Janke et al, 2004). The BY4742 deletion strain for *GID4* was taken from a yeast deletion library (Giaever et al, 2002). Yeast strains used in this study are listed in Table S5.

For yeast transformation, the lithium acetate method was used (Gietz et al, 1992). Oxa1-Ura3 and $b_2\Delta_{19(1-167)}$-DHFR plasmids were described previously (Hansen et al, 2018; Boos et al, 2019). Empty plasmids were used for control.

Yeast cells were grown in a yeast full medium containing 1% (wt/vol) yeast extract and 2% (wt/vol) peptone at 30°C. As carbon sources, 2% glucose (YPD), 2% galactose (YPGal), 2% glycerol (YPG), or 2.7% lactate (YPLac) was used. The strains carrying plasmids were grown in the minimal synthetic medium containing 0.67% (wt/vol) yeast nitrogen base and 2% glucose (SD), galactose (SGal), or lactate (SLac), as indicated. For plates, 2% agar was added to the medium. To induce protein expression from the *GAL1* promoter, 0.5% galactose was added to yeast cultures.

## Growth assays and viability test

For drop dilution assays, respective yeast strains were grown in yeast full or synthetic media to the mid-log phase. After harvesting 0.5 $OD_{600}$ of cells and washing with sterile water, a 1:10 dilution series was prepared. 3 $\mu$l of each dilution was dropped on agar plates containing full or synthetic media. Pictures of the plates were taken after different days of incubation at 30°C.

For testing growth in liquid media, growth curves were performed in 96-well plates in technical triplicates. The ELx808 Absorbance Microplate Reader (BioTek) was used for automated OD measurement at 600 nm. With starting at an $OD_{600}$ of 0.1, $OD_{600}$ was measured every 10 min for 72 h at 30°C.

To test the viability of cells, respective yeast strains were inoculated in YPD medium, diluted once, and then continuously grown in YPD for 10 d at 30°C. Samples were taken every day. For this, 1 $OD_{600}$ of cells was harvested and washed with sterile water, and 100 $\mu$l of $OD_{600}$ of 0.01 was plated out on YPD agar plates. After incubating for one to 2 d at 30°C, colony numbers were counted.

## Halo assay for CCCP sensitivity

For the halo assay, yeast strains were precultured in the YPG medium. After harvesting 1 $OD_{600}$ of cells and washing with sterile water, 100 $\mu$l of $OD_{600}$ of 0.01 was plated out on glycerol plates. A filter plate with 10 $\mu$l of CCCP (10 mM) was placed in the middle of

the plate. After incubation for 1 d at 30°C, halo areas were measured. Filter plates with 10 $\mu$l DMSO served as negative controls.

## Preparation of whole-cell lysates

4 $OD_{600}$ of yeast cells were harvested and washed with sterile water. Pellets were resuspended in 40 $\mu$l/$OD_{600}$ Laemmli buffer containing 50 mM DTT. Cells were lysed using a FastPrep-24 5 G homogenizer (MP Biomedicals) with three cycles of 20 s, speed 6.0 m/s, 120-s breaks, and glass beads (∅ 0.5 mm) at 4°C. The lysates were heated at 96°C for 5 min and stored at −20°C until visualization by Western blotting.

## Antibodies

Antibodies were raised in rabbits using recombinant purified proteins. The secondary antibody was obtained from Bio-Rad (Goat Anti-Rabbit IgG (H+L)-HRP Conjugate, #172-1019). The horseradish peroxidase–coupled HA antibody was purchased from Roche (Anti-HA-Peroxidase, High Affinity [3F10], #12 013 819 001). Antibodies were diluted in 5% (wt/vol) nonfat dry milk in 1× TBS buffer with the following dilutions: anti-HA 1:500 and anti-rabbit 1:10,000. Antibodies used in this study are listed in Table S6.

## CCCP treatment

Analysis of Mdj1 and Rip1 precursor accumulation upon carbonyl cyanide m-chlorophenyl hydrazone (CCCP) treatment was performed as described before (Mårtensson et al, 2019).

## Tom22 import assay

The mitochondrial membrane insertion of Tom22 was analyzed using [$^{35}$S]Tom22 with three additional methionine residues at the C-terminus as previously described (Ellenrieder et al, 2016). For each import reaction, 10 $\mu$l of lysate was mixed with 50 $\mu$g of mitochondria in the import reaction buffer (3% [wt/vol] bovine serum albumin, 250 mM sucrose, 80 mM KCl, 5 mM $MgCl_2$, 2 mM $KH_2PO_4$ and 10 mM MOPS-KOH, pH 7.2, 2 mM ATP, 2 mM NADH, 12 mM creatine phosphate, and 100 $\mu$g/ml creatine kinase). The import was performed at 25°C and stopped by transfer on ice. Samples were treated with proteinase K (80 $\mu$g/ml) for 15 min on ice. The protease was inactivated by the addition of 20 $\mu$M PMSF. To monitor the assembly of Tom22, mitochondria were lysed with 1% [wt/vol] digitonin lysis buffer (20 mM Tris–HCl, pH 7.4; 0.1 mM EDTA; 50 mM NaCl; and 10% [vol/vol] glycerol) after import of [$^{35}$S]Tom22. Protein complexes were separated by blue native gel electrophoresis (Priesnitz et al, 2020).

## Fermentor measurements

To measure population-synchronized metabolic oscillations, yeast cultures were stably synchronized with respect to the yeast metabolic cycle as previously described (Tu et al, 2005; Amponsah et al, 2021). All experiments were performed in a CEN.PK113-1A strain background. A Biostat A fermentor (Sartorius Stedim Systems), with a culture volume of 800 ml, was used for all experiments. Culture

media consisted of 10 g/l glucose, 1 g/l yeast extract, 5 g/l $(NH_4)_2SO_4$, 2 g/l $KH_2PO_4$, 0.5 g/l $MgSO_4$, 0.1 g/l $CaCl_2$, 0.02 g/l $FeSO_4$, 0.01 g/l $ZnSO_4$, 0.005 g/l $CuSO_4$, 0.001 g/l $MnCl_2$, 2.5 ml 70% $H_2SO_4$, and 0.5 ml/l Antifoam 204.

Fermentor runs were initiated by the addition of a 20-ml pre-culture, which was grown to stationary phase in YPD medium at 30°C. The fermentor was run in batch-culture mode at 30°C, with an aeration rate of 1 liter/min and constant stirring at 530 rpm. A constant pH of 3.4 was maintained by the automated addition of 10% (wt/vol) NaOH. Fermentor cultures were grown until ~5 h after the exhaustion of carbon source as determined by continuous monitoring of culture oxygen saturation. Subsequently, a continuous culture was initiated by the addition of fresh media to the culture vessel at a dilution rate of 0.05 $h^{-1}$. Culture oxygen saturation was automatically recorded with a sampling interval of 10 s.

## RNA isolation

For RNA isolation, the RNeasy kit (Qiagen) and the RNase-Free DNase Set (Qiagen) were used. The cells were grown to the mid-log phase. 4 OD of cells were harvested and lysed using the FastPrep-24 5 G homogenizer (MP Biomedicals) with three cycles of 20 s, speed 6.0 m $s^{-1}$, and 120-s breaks. RNA purity and concentration were tested using a NanoDrop fluorometer.

## Quantitative real-time PCR (qRT-PCR) assays

qRT-PCRs were performed in technical triplicates using a CFX96 Touch Real-Time PCR Detection System (Bio-Rad). For cDNA synthesis and subsequent qRT-PCR, the Luna Universal One-Step RT-qRT-PCR kit (NEB) was used. For normalization, *ACT1* was selected as a housekeeping gene. Primers used for qRT-PCR are listed in Table S7. The primer efficiency was tested by generating standard curves for cDNA serial dilutions using the iQ SYBR Green Supermix kit (Bio-Rad). Only primer pairs with efficiencies between 90% and 110% were used. The 2–ΔΔCt method (Livak & Schmittgen, 2001) was used for data analyses, normalizing the gene expression values to the housekeeping gene transcript levels and to the wild-type or empty vector strain as reference. Statistical significance was assessed using a *t* test.

## Sample preparation and mass spectrometric identification of proteins

For dynamic SILAC mass spectrometry (MS), YPH499 Δarg4 wild-type and Δubc8 cells were cultured in the SLac medium containing light [$^{14}N_2$, $^{12}C_6$]-lysine and [$^{14}N_4$, $^{12}C_6$]-arginine isotopes. The cells were diluted continuously to keep them in the exponential growth phase while increasing the culture volume stepwise up to 300 ml. For time point t(0), 100 ml of each culture was harvested (5,000g, 5 min, RT), washed with sterile water, and shock-frozen with liquid nitrogen. Samples were stored at –80°C for further analysis. To analyze the metabolic shift (Figs 2 and S2), 2 × 100 ml of every culture was harvested (5,000g, 5 min, RT) and washed with 30 ml SLac medium without lysine and arginine (5,000g, 5 min, RT). The cells were resuspended in 200 ml SLac+2% glucose or SLac as control. Both media only contained the heavy amino acid isotopes of lysine ($^{15}N_2$,

$^{13}C_6$) and arginine ($^{15}N_4$, $^{13}C_6$). Cultures were incubated at 30°C and 140 rpm for one doubling time of the cells. Cell growth was continuously monitored by measuring $OD_{600}$. Samples (t(1) Gluc and t(1) Lac) were harvested and treated as described before for the samples t(0).

MS samples were prepared according to a published protocol with minor adaptations (Kulak et al, 2014). Cell lysates from 25 OD were prepared in 100 µl lysis buffer (6 M guanidinium chloride, 10 mM TCEP-HCl, 40 mM chloroacetamide, and 100 mM Tris, pH 8.5) using a FastPrep-24 5 G homogenizer (MP Biomedicals) with three cycles of 20 s, speed 8.0 m/s, 120-s breaks, and glass beads (∅ 0.5 mm) at 4°C. Samples were heated for 10 min at 96°C and afterward centrifuged twice for 5 min at 16,000g. In between, the supernatant was transferred to fresh Eppendorf tubes to remove all remaining glass beads. Protein concentrations were measured using the Pierce BCA Protein Assay (#23225; Thermo Fisher Scientific). For protein digestion, 36 µg of protein was diluted 1:10 with LT-digestion buffer (10% acetonitrile and 25 mM Tris, pH 8.8). Trypsin (#T6567; Sigma-Aldrich) and Lys-C (#125-05061; Wako) were added to the samples (1:50 w/w). Samples were incubated overnight at 37°C and 700 rpm. After 16 h, fresh trypsin (1:100 w/w) was added for 30 min (37°C, 700 rpm). The pH of samples was adjusted to < 2 with TFA (10%), and samples were centrifuged for 3 min at 16,000g and RT. Desalting/mixed-phase cleanup was performed with three-layer SDB-RPS stage tips (Cat. no. 2241). Samples were dried down in SpeedVac and resolubilized in 12 µl buffer A++ (buffer A [0.1% formic acid] and buffer A* [2% acetonitrile and 0.1% trifluoroacetic acid] in a ratio of 9:1).

To analyze the metabolic shift by label-free mass spectrometry (Figs 4B and S5B), the respective yeast strains were cultured in YPG. Glucose (final concentration: 2%) was added to cultures for 4 h. To analyze proteomes of continuously grown cultures by label-free mass spectrometry, the respective strains were cultured in YPD, YPGal, or YPLac. For both experiments, 10 $OD_{600}$ of cells were harvested (5,000g, 5 min, RT) and washed with sterile water. Pellets were snap-frozen in liquid nitrogen and stored at –80°C for further analysis. Samples were prepared for mass spectrometry as described above with minor changes: Cell lysates of 10 OD were prepared in 200 µl lysis buffer, and 25 µg of protein was used for trypsin and Lys-C digestion.

For both MS experiments, the digested peptides were separated on reversed-phase columns (50 cm, 75 µm inner diameter) packed in-house with C18 resin ReproSil-Pur 120, 1.9 µm diameter (Dr. Maisch) using an Easy-nLC 1200 system (Thermo Fisher Scientific) directly coupled to a Q Exactive HF mass spectrometer (Thermo Fisher Scientific). A 3-h gradient from 5 to 95% Solvent B (Solvent A: aqueous 0.1% formic acid; Solvent B: 80% acetonitrile and 0.1% formic acid) at a constant flow rate of 250 nl/min was used to elute bound peptides. Furthermore, details of the gradient and instrument parameters are provided with the raw files uploaded to the ProteomeXchange repository.

MS data were processed using the MaxQuant software (version 1.6.10.43) (Cox & Mann, 2008; Cox et al, 2011; Tyanova et al, 2016) and a *Saccharomyces cerevisiae* proteome database obtained from UniProt.

## Quantification and statistical analysis

MaxQuant output files were processed using Perseus and the R programming language. Each condition was measured in four

replicates. For the lactate-to-glucose shift experiment (Fig 2), the proteins were filtered to contain at least three valid values in at least one of the compared conditions. Log$_2$ protein intensities (combined intensity of the heavy and light channels) were mean-centered. For the dynamic SILAC experiment (Fig 2E), the same normalization factors were applied to the individual light and heavy channels. To compare conditions, a t test was performed and P-values were adjusted for multiple testing (Benjamini & Hochberg, 1995). Hierarchical clustering (Fig 2E) was carried out in Perseus. Mean-centered log$_2$ protein intensities were filtered using a multiple-sample ANOVA test (FDR > 0.01, S0 = 1) implemented in Perseus, and Z-scored protein intensities were clustered according to the Euclidean distance. The heatmap was visualized in R using the pheatmap package. Gene set enrichment analysis was performed using Fisher's exact test. P-values of gene set enrichments were adjusted for multiple hypothesis testing using the Benjamini-Hochberg procedure (Benjamini & Hochberg, 1995).

For the label-free experiment (Figs 4 and S4), the proteins were filtered as described above and label-free quantitation protein intensities were cleaned for batch effects using limma (Ritchie et al, 2015) and further normalized using variance stabilization-normalization (Huber et al, 2002). The proteins were tested for differential expression using the limma package, and P-values were adjusted for multiple hypothesis testing using the Benjamini-Hochberg procedure (Benjamini & Hochberg, 1995).

Western blot analyses were independently replicated with similar results, and representative data are shown in the figures. Quantification was performed with Fiji/ImageJ, and significance testing was performed with a t test.

### Sample preparation and mass spectrometric measurement of methylglyoxal (MG) levels in yeast

Respective yeast strains were cultured in a yeast full medium containing 0.2% glucose until the mid-log phase at 30°C. Glucose (final concentration: 2%) was added to cultures for 1 h. For mass spectrometric analysis, 50 OD$_{600}$ of cells were harvested (5,000$g$, 5 min, RT). The cell pellets were washed with sterile water twice (5,000$g$, 5 min, RT; and 16,000$g$, 2 min, RT, respectively) and stored at −80°C for further analysis. MG levels were determined by mass spectrometry after derivatization to 2-methylquinoxaline (2-MQ). For sample derivatization, preparation of standards, and mass spectrometric measurement, the protocol from Rabbani and Thornalley was adjusted for yeast cells (Rabbani & Thornalley, 2014). In detail, the cells were resuspended in 300 $\mu$l ice-cold tri-chloroacetic acid (TCA, 20% [wt/vol] with 0.9% [wt/vol] sodium chloride). Glass bead lysis was performed using a FastPrep-24 5G (MP Biomedicals) with three cycles of 20 s, speed 8.0 m/s, 120-s breaks, and glass beads (∅ 0.5 mm) at 4°C. Samples were centrifuged (10,000$g$, 10 min, 4°C), and 140 $\mu$l of the supernatant was used for derivatization. Sample derivatization was performed as described by Rabbani and Thornalley, but a quadruplicate volume of chemicals and d$_4$-2-methylquinoxaline (d$_4$-2-MQ) instead of iso-topically labeled MG was used. Nine standards with different amounts of MG were prepared for a calibration curve (0–1,091 nM).

For analysis, an Agilent 1200 HPLC system (Agilent Technologies) coupled with an API 3200 tandem mass spectrometer (AB Sciex) was used. Separations were performed using a RP 18 column (XBridge, C18, 2.5 $\mu$m, 2.1 × 50 mm; Waters). The injection volume was 15 $\mu$l, and the used flow was 250 $\mu$l/min. The mobile phases consisted of water with 0.1% TFA (A) and a mixture of water with acetonitrile (50/50 [vol/vol]) with 0.1% TFA (B). The concentration of B was 0% at 0 min and increased to 100% for 5 min and is held at 100% for 5 min, followed by a reconditioning step. The measurement was carried out in the multiple reaction monitoring mode with positive ioni-zation. An electrospray ionization (ESI) source was used with source parameters as follows: ion spray voltage, 5,000 V; temperature, 650°C; nebulizer gas, 65 $\psi$; and heater gas, 65 $\psi$. The characteristic combinations of the parent ions and the product ions (Q1→Q3, $m/z$) for 2-MQ and d$_4$-2-MQ were 145.1→77.1*, 145.1→92.1 and 149.1→81.1*, 149.1→122.1, respectively. For quantification, a calibration curve with a peak area ratio of 2-MQ/d$_4$-2-MQ, where the transitions marked with an asterisk were used, against the amount of MG (pmol) was created. The MS data were evaluated by Analyst version 1.7.2 (AB Sciex). From a linear regression model, the amount of MG was deduced. The limit of detection and limit of quantification for the described method are 3.5 and 7 nM 2-MQ, respectively.

### Miscellaneous

The following experiments were performed as published before: isolation of mitochondria and in vitro import of radiolabeled proteins (Peleh et al, 2015), fluorescence microscopy (Westermann & Neupert, 2000), blue native gel electrophoresis (Priesnitz et al, 2020), and the proteasome-associated control element reporter assay (Boos et al, 2019).

# Data Availability

The mass spectrometric proteomics data (see also Tables S1, S3, and S4) have been deposited to the ProteomeXchange Consortium via the PRIDE (Perez-Riverol et al, 2019) partner repository with the dataset identifier PXD033171 (dynamic SILAC), PXD033193 (label-free dataset), and PXD036508 (continuously grown cultures).

# Supplementary Information

# Acknowledgements

We thank Lena Krämer, Anna Gröger, Simone Stegmüller, Sabine Knaus, Andrea Trinkaus, and Ralph Mahlberg for technical assistance and Martin van der Laan, Chris Meisinger, Klaus Pfanner, and Peter Kötter for reagents; and Zuzana Storchova for support. The contribution of Katja Hansen, Naama Aviram, and Maya Schuldiner was essential for the identification of Ubc8 as a factor relevant for mitochondrial protein biogenesis. This project was funded by grants from the Deutsche Forschungsgemeinschaft (HE2803/10-1 and RTG 2737 STRESSistance to JM Herrmann, and SFB1218 (project ID 269925409) to T Becker) and the Landesschwerpunkt BioComp.

## Author Contributions

S Rödl: conceptualization, data curation, validation, investigation, visualization, methodology, and writing—original draft, review, and editing.

F den Brave: conceptualization, data curation, investigation, and writing—review and editing.

M Räschle: conceptualization, data curation, investigation, methodology, and writing—review and editing.

B Kizmaz: data curation, investigation, methodology, and writing—review and editing.

S Lenhard: data curation, investigation, and writing—review and editing.

C Groh: data curation, formal analysis, and writing—review and editing.

H Becker: data curation, investigation, and writing—review and editing.

J Zimmermann: data curation, methodology, and writing—review and editing.

B Morgan: conceptualization, data curation, methodology, and writing—review and editing.

E Richling: conceptualization, formal analysis, and writing—review and editing.

T Becker: conceptualization, supervision, investigation, and writing—review and editing.

JM Herrmann: conceptualization, data curation, supervision, validation, investigation, methodology, writing—original draft, and project administration.

## Conflict of Interest Statement

The authors declare that they have no conflict of interest.

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
