## [Reviewer comments · Life Science Alliance]

Life Science Alliance

The metabolite-controlled ubiquitin conjugase Ubc8 promotes mitochondrial protein import

Saskia Rödl, Fabian den Brave, Markus Räsche, Büsra Kizmaz, Svenja Lenhard, Carina Groh, Hanna Becker, Jannik Zimmermann, Bruce Morgan, Elke Richling, Thomas Becker, and Johannes Herrmann

DOI: <https://doi.org/10.26508/lsa.202201526>

Corresponding author(s): Johannes Herrmann, University of Kaiserslautern

Review Timeline:

Submission Date:	2022-05-16
Editorial Decision:	2022-06-17
Revision Received:	2022-09-09
Editorial Decision:	2022-09-26
Revision Received:	2022-09-30
Accepted:	2022-09-30

Scientific Editor: Novella Guidi

Transaction Report:

June 17, 2022

Re: Life Science Alliance manuscript #LSA-2022-01526-T

Dr. Johannes M Herrmann
University of Kaiserslautern
Cell Biology
Erwin-Schroedinger-Strasse 13
Kaiserslautern D-67663
Germany

Dear Dr. Herrmann,

Thank you for submitting your manuscript entitled "The metabolite-controlled ubiquitin conjugase Ubc8 regulates mitochondrial protein import" to Life Science Alliance. The manuscript was assessed by expert reviewers, whose comments are appended to this letter. We invite you to submit a revised manuscript addressing the Reviewer comments.

Thank you for this interesting contribution to Life Science Alliance. We are looking forward to receiving your revised manuscript.

Sincerely,

B. MANUSCRIPT ORGANIZATION AND FORMATTING:

Reviewer #1 (Comments to the Authors (Required)):

The transition of yeast cells from respiratory to fermentative growth requires reprogramming of cellular and mitochondrial metabolism. A key factor mediating this metabolic switch is the ubiquitin ligase Ubc8, a central component of the glucose induced degradation deficient (GID) complex that promotes turnover of gluconeogenesis enzymes when glucose becomes available. Here, Rödel et al. describe a second, unexpected function for Ubc8 in maintaining mitochondrial protein import. Using a combination of genetic screening and dynamic SILAC labelling, the authors find that loss of Ubc8 not only blocks the degradation of gluconeogenesis enzymes upon respiratory-to-fermentative shift but also leads to cytoplasmic retention of mitochondrial preproteins and thus impaired mitochondrial function. Mechanistically, the authors find that Ubc8 selectively promotes incorporation of the import receptor Tom22 into the translocase of the outer membrane (TOM) complex, a central gate for protein import into mitochondria. Consequently, deletion of Ubc8 impairs Tom22-dependent mitochondrial protein biogenesis, delays the metabolic transition of yeast cells to fermentative growth, and cause increased lethality during the stationary phase of glucose grown cells.

Overall, the paper is well written and the data confirming the previously described role of Ubc8 in catabolite degradation upon transition to fermentative growth as well as its new function in Tom22 protein biogenesis are convincing. However, the extent to which deletion of Ubc8 impairs global mitochondrial protein import causing precursor accumulation in the cytoplasm (as stated in the manuscript) and mitochondrial remodelling during metabolic transitions appears less clear. In addition, if mitochondrial function is indeed impaired in Δ ubc8 cells remains unclear due to several discrepancies within the manuscript. Therefore, while Ubc8 appears to affect mitochondrial protein dynamics, the paper in its current state goes a bit too far in the interpretation of the available data. As such, some revisions to either provide sufficient data or change the text to better fit the available results would be necessary.

Major comments:

1. The authors show that deletion of Ubc8 increases the cytosolic levels of two artificial mitochondrial fusion proteins: Oxa1-Ura3, which was used to identify yeast mutants (including Ubc8) that slow mitochondrial protein import (this study and Hansen et al., 2018), and b2(1-167) Δ 19-DHFR, a fusion construct widely used to assess mitochondrial protein import in vitro. Based on these results, the authors conclude that loss of Ubc8 causes "increased amounts of mitochondrial precursor proteins in the cytosol". As it stands, the data presented in the manuscript do not sufficiently support this conclusion:

First, while b2(1-167) Δ 19-DHFR levels are indeed elevated in Δ ubc8 cells it remains unclear if this pool is cytosolic or mitochondrial. Moreover, whether the observed delay in b2(1-167) Δ 19-DHFR turnover in Δ ubc8 cells is due to impaired degradation in the cytosol or within the mitochondrial matrix is not assessed. Of note, the immunoblot (Figure 1D-E) at best shows a minimal reduction in b2(1-167) Δ 19-DHFR turnover in absence of Ubc8 and the quantification only significantly differs at a single timepoint.

Second, whether deletion of Ubc8 indeed causes global retention of mitochondrial proteins within the cytosol is not assessed. This is surprising, as the authors subsequently use proteome analysis to analyze differences in the cellular proteome of wild-type (WT) and Δ ubc8 cells upon shift from lactate to glucose. The authors should at the very minimum assess the accumulation of mitochondrial protein precursors in the cytosol using subcellular fractionation and/or Western Blot analysis. Unprocessed mitochondrial preproteins should be readily detectable by Western Blot

Third, in Figures 2E and 3A-D the authors demonstrate Ubc8-dependent degradation of gluconeogenetic enzymes upon switch to fermentative growth. Surprisingly, the effect of Ubc8 on mitochondrial protein turnover is not further elucidated. Therefore, whether Ubc8 "promotes the biogenesis of mitochondrial proteins" as claimed or has a (potentially additional) degradation-based effect remains unclear. Since the data is already available, the authors should include this analysis in an updated version of the manuscript and should also assess whether any observed decline in mitochondrial protein levels is sensitive to cytosolic proteasome inhibition.

2. The authors convincingly show that deletion of Ubc8 impairs cell survival in glucose-grown cultures under stationary conditions (Figures 3H and S3B), impairs metabolic adaptation upon respiratory-to-fermentative transition, and demonstrate an overall reduction in mitochondrial protein levels (Figure 2D) in Δ ubc8 cells. Based on these results, the authors infer a "reduced functionality of mitochondria in Δ ubc8 cells". However, a clear link between mitochondrial dysfunction in Δ ubc8 cells and the observed metabolic defects remains missing. Notably, Δ ubc8 cells exhibit no growth defects on respiratory carbon sources (Figure S1E) suggesting that mitochondria are indeed fully functional in absence of Ubc8. This latter discrepancy may be partially explained by the fact that the effect of Ubc8 on Tom22 levels "was less pronounced in glycerol grown cells".

Unfortunately, biochemical data supporting this conclusion is not provided in the manuscript. This discrepancy needs to be addressed. Along those lines, the authors should test if temperature sensitive alleles of Tom22 phenocopy the observed metabolic effects of Δ ubc8. This would allow the authors to uncouple the mitochondrial function of Ubc8 (Tom22 biogenesis) from its role in the GID complex.

3. The authors demonstrate that Ubc8 is required for maintaining mitochondrial Tom22 levels and that deletion of Tom22 impairs its incorporation into the TOM complex. However, whether this block occurs at the step of complex assembly (as concluded by the authors) or earlier (i.e. at insertion into the outer membrane) remains unclear and should be addressed. Of note, the levels of the SAM-Mdm10 complex which mediates Tom22 assembly are reduced in Figure 5A, but the authors state that SAM levels are unaltered in Δ ubc8 cells. This should be changed. Along those lines, it seems surprising that despite the strong reduction in Tom22 and other mitochondrial protein levels (noted in Figure 2D), as well as the claimed import defect caused by the lack of Tom22, all other mitochondrial proteins shown in the blot appear to be unaffected by Ubc8 knockout. Please add text to address this discrepancy.

4. In the discussion the authors speculated that Ubc8 may mediate its effect on Tom22 levels through degradation of a cytosolic phosphatase or kinase that antagonizes Tom22 import. However, as Figure 5B shows reduced Tom22 assembly into TOM in vitro using radiolabelled Tom22 and isolated mitochondria without the presence of cytosolic enzymes, this seems counterintuitive. Please add supporting data or adjust the manuscript text accordingly

5. If possible, please include a comparison of the proteome of wildtype and Δ ubc8 cells continuously grown in glucose.

Minor comments:

In Figure S3A the authors assess MGO levels after 1h of growth on glucose. However, growth differences between wildtype and Δ ubc8 only become apparent after 20-48h. Do MGO levels remain unchanged in Δ ubc8 cells at later timepoints? This analysis is critical to conclude that toxic metabolites do not contribute to increased lethality in the stationary phase. In case it is not feasible to include these data, please adjust the conclusions accordingly.

The authors assess import of Pet9 by SDS-PAGE and conclude its biogenesis is impaired in Δ ubc8 cells (Figure 4A). However, upon import mitochondrial carriers assemble into multi-subunit complexes and thus, their biogenesis is best assessed by BN-PAGE. Please include this analysis.

The increase in the Oxa1 precursor form (Figure 4A) points to additional import defects after the precursor has been translocated across the outer membrane. Please address this in the manuscript text.

The manuscript states that Δ ubc8 cell growth rate lags behind following (but not before) the diauxic shift. It would be helpful to include language or data showing how the timepoint of the diauxic shift was determined.

For several figures, most prominently the immunoblots, data on the number of repetitions should be included.

Several Figures and Figure Legends would benefit from additional or improved labelling:

1. Figure 1C is lacking a figure legend to distinguish the empty vector controls
2. Please annotate the other well-growing strain shown on the plate in Fig S1B (likely Ema35?) and mention it in the manuscript text.
3. Figure S1D is not addressed or referenced in the text at all.
4. Information about the employed statistical tests and significance levels is missing in Figure 1E, as well as in the proteomics data (Figures 2C, 4B, S2A and S4B). Please clarify which data points reflect significant outliers.
5. In Figure 2C, the annotation of the gluconeogenic proteins is difficult to see and the annotation of Jen1 in the first plot is not explained and missing in the second plot.
6. Figure S2A is lacking the names of mitochondrial proteins referenced in the figure description.
7. Figure 2D: Please consider moving Figure S2B to the main Figure as the data is very convincing.
8. In Figure 2D, please mark mitochondrial proteins with a line next to the plot instead of referring to a vaguely to "many" mitochondrial proteins. Consider separating these out in an independent heat map.
9. In Figure 2D it is not clear whether "protein levels increased/reduced in Δ ubc8" refers to the t0 lactate state, the t1 shifted cultures or both.
10. In Figure 2F, comparisons of mitochondrial proteins with other compartments such as ER or peroxisomes would be helpful to assess whether Ubc8 selectively affects the mitochondrial proteome.
11. Please explain what the additional Icl1 band is that appears in Δ ubc8 cells (Figure 3C).
12. It would be helpful to include numerical/statistical data on the exact timepoint of when WT and Δ ubc8 cells enter oscillation and the cycle time should be included in the manuscript and/or the figure (Figure 3G)
13. The figure description of Figure 3H describes that "cultures were grown to mid-log phase in glucose medium" while the text refers to a switch "from glycerol medium", this should be corrected.

Reviewer #2 (Comments to the Authors (Required)):

In this study, the authors investigated the role of the ubiquitin conjugase Ubc8 as a metabolism- dependent remodeler in yeast. Ubc8 is a conserved and well-studied component of the GID (Glucose Induced degradation Deficient) complex, which forms the catabolite control system upon transition from non-fermentative to fermentative growth medium.

In a previously published screen, the authors have found that Ubc8 is required for efficient import of the model substrate Oxa1-Ura3 into mitochondria. The authors confirm this finding and use a series of experiments to demonstrate that Ubc8 is required to maintain Wildtype-like levels of Tom22, normal assembly of the TOM complex and efficient import of proteins into mitochondria.

Cells must constantly be able to adapt to changing metabolic conditions, the underlying mechanisms, however, remain incompletely understood. The present study therefore addresses an important cell biological question and provides new insights by linking Ubc8 to mitochondrial import and biogenesis. The presented experiments are of a high technical quality and acceptance of this manuscript is recommended. However, a few points should be addressed in a revised version of the manuscript:

Figure 1C would benefit from a legend indicating, which color corresponds to what strain. Since the 'WT' label is currently in between three lines in the graph it is slightly ambiguous. Upon comparing Fig S1A and S1C, it seems as if the WT+ Δ N-Oxa1-Ura3 strain grows on medium lacking Uracil on the drop dilutions (Fig. S1A), but not in the growth curve (Fig S1C). However, this could be due to my misinterpretation of the colors in Figure S1C, which could be changed to make them easier to distinguish. While evident for an expert reader, the purpose of the Δ N-Oxa1-Ura3 construct could be briefly explained.

The authors should discuss the data presented in Figure 1E in more detail. Why is there an effect on Su9-DHFR in the first 30 minutes, but not after that in Δ ubc8 cells? Removal of the remaining precursor even seems to speed up after the initial 30 min.

The experiment presented in Figure 1F should be performed in triplicates.

The oscillatory oxygen consumption (Figure 3F) should be explained in slightly more detail, which would be helpful to the reader. How can the oscillation of oxygen consumption be explained (even for WT)? Along the same lines, it is not clear to me why the shorter cycles in Δ ubc8 cells are 'consistent with a disrupted ability to switch between different metabolic states'.

The text should briefly explain the potential toxic effects and origin of methylglyoxal.

Ubc8 is intimately linked to adaptation of cells to changing metabolic requirements. The authors of the present study confirm this by demonstrating that loss of Ubc8 leads to impaired removal of Gluconeogenesis enzymes upon shift from respiratory to fermentative growth. The authors show clearly that deletion of UBC8 affects accumulation of Tom22 and compromises mitochondrial import. Whether these effects can also be linked to a metabolic shift and whether this is a conclusion that the authors draw, however, remains less clear to me at this stage. The screen presented in Figure 1 was performed on Glucose, I assume, and thus deletion of UBC8 leads to defective import and accumulation of Oxa1-Ura3 on a fermentable carbon source. Clearance of Su9-DHFR precursor is compromised during a shift from Galactose to Glucose in Figure 1D. Tom22 appears decreased in Δ ubc8 cells in cells grown on glycerol (Figure S4B) and upon shift from Glycerol to Glucose medium (Figure 4B). Thus an effect on mitochondrial protein import is somewhat apparent under all tested conditions. The authors should try to clarify if Ubc8 is important for mitochondrial import and TOM complex assembly generally or only upon metabolic shifts. In particular, it would be important to assess Tom22 protein levels in WT and Δ ubc8 strains by Western Blot in cells continuously grown on YPG. Is the result different from the data presented in Figure 4C? In the current form of the manuscript, it is difficult to tell whether there is a difference regarding Tom22 levels between cells grown in Glycerol or cells shifted from Glycerol to Glucose by comparing figures 4B and S4B. Alternatively, Ubc8-dependent Tom22 regulation upon a metabolic shift could be determined by collecting multiple samples over the course of a shift from glycerol to glucose in WT and Δ ubc8 cells. Similarly, in organello import experiments could be performed with mitochondria isolated from glycerol or after a shift onto glucose. Along these lines, it should be stated for Figure 4A from which growth medium mitochondria were isolated to perform the import.

A reduction of Tom22 protein levels in Δ ubc8 cells could have several reasons. While Figure 5B suggests that Tom22 accumulation is caused by compromised assembly into the TOM complex, a potential transcriptional response in the absence of Ubc8, should be excluded by performing RT-qPCR experiments to assess the Tom22 mRNA levels in WT and Δ ubc8 cells grown in glycerol or upon a shift from glycerol to glucose.

A brief discussion about the change in the TOM complex size in Δ ubc8 (Figure 5A) would be helpful. What change could lead to this size shift? Figure 5C suggests that it could be a conversion of the dimeric into a monomeric version of the TOM complex, this however seems unlikely due to the rather small shift.

Figure 5B shows that in organello assembly of radiolabeled Tom22 into the TOM complex is compromised. In this experiment, cytosolic components should be absent from the reaction. Therefore, it seems unlikely that cytosolic kinases or phosphatases play a direct role on Tom22 for its assembly. This point should be discussed.

The text refers to Figure S4C by stating that absence of Ubc8 leads to 'wide-ranging depletion of mitochondrial proteins, in particular those of the matrix for which the Tom22 levels seem particularly important.'. It may be the case that I am not an expert in the interpretation of violin-plots, but to me it does not appear clear where in the plot I would find support for this statement. Perhaps the figure could be improved by highlighting the parts of the plot that support this conclusion. Further information could be given in the legend.

Minor points:

The authors should indicate the precursor and mature forms of DHFR in Figure 1D.

Figure 3H legend - "cultures were grown to mid-log phase in glucose medium..." is inconsistent with the main text on page 9 ("Next we tested the growth of cells after switching from glycerol medium to different concentrations of glucose"). This should be rectified by the authors.

Typo: Reference to Figure 3G in the text says 'read arrows' instead of 'red arrows'. It should be added to the legend of the figure, what the red arrows exactly point at.

Legend Figure S2 states that 'Names of significantly changed mitochondrial proteins are indicated'. This is not the case and should be corrected.

No real explanation is given for Figure S1D. What is the relevance of the comparison of Ubc8 to other proteins?

Reviewer #3 (Comments to the Authors (Required)):

Performing a genome-wide screen for gene modifications affecting mitochondrial protein import efficiency, the authors identify UBC8, a E2 ubiquitin conjugase within the conserved GID complex. Previous work has shown that the Ubc8/GID complex regulates the degradation of gluconeogenic enzymes upon metabolic shift from non-fermentable to fermentable carbon sources, a process termed catabolite degradation. In this manuscript, the authors show that the absence of Ubc8 affects mitochondrial protein import, mitochondrial protein levels, and in particular results in low Tom22 protein levels with consequence for TOM complex assembly. Thus, the authors propose that Ubc8 positively affects Tom22/TOM functions ensuring efficient import and steady state levels of mitochondrial proteins.

This work uncovers an unexpected link between the Ubc8/GID complex and mitochondrial protein homeostasis. The manuscript is well written and data quality is overall high. However, the manuscript displays a few weaknesses that should be addressed.

The authors use a number of different growth conditions, which make it difficult/confusing for the reader to compare results/experiments with each other. Conditions are not always indicated in the figure legends, and it is often not explained why the authors use a particular condition for certain experiments.

The authors propose that Ubc8 is a positive regulator of Tom22 assembly in TOM complex and mitochondrial import. However, the Ubc8/GID complex is activated when glucose is present, which represses mitochondrial biogenesis. How do the authors explain this counterintuitive connection? Is there a particular subset of mitochondrial proteins that is affected by the potential Ubc8-Tom22 regulatory arm, which could aid in mitochondrial adaptation to altered carbon/growth conditions?

The authors describe a novel and interesting link between the GID complex and TOM complex assembly with clear effects on Tom22. However, the molecular underpinning of this connection remains elusive. Specifically, it is unclear whether the GID complex directly affects Tom22 or other regulators of Tom22/TOM assembly. Given their expertise in proteomics, would the authors be able to define ubiquitination patterns by reanalyzing their whole cell proteomics data to determine potential effects of Ubc8 on candidate proteins?

Given that the molecular mechanisms remain elusive, the term "regulate" in the title seems to be a bit of an overstatement at this point. I am aware that "regulate" is often (over)used instead of the more appropriate "affect". To me, however, "regulate" implies a direct regulatory role, for example, direct Ubc8-dependent ubiquitination of Tom22 or its assembly factors. This, however, has not been demonstrated, yet.

Along these lines, does Ubc8 overexpression increase Tom22 assembly into the TOM complex and enhance protein import into mitochondria?

The effect of *ubc8* on Tom22 and TOM complex assembly could explain the observed changes on mitochondrial proteins. However, the authors do not provide experimental evidence whether this sufficient to explain the defects observed in the absence of Ubc8. For example, does TOM22 overexpression (at least partially) rescue (some of) the *ubc8*-associated phenotypes?

Based on figures 1D and E, the authors conclude that the DHFR-reporter is imported significantly more slowly in *ubc8* mutants compared with WT cells. However, only one data point (30 min) shows statistical significance. Thus, the effects on mitochondrial import are rather minor. Why do the authors shift cells from galactose to lactate? Wouldn't the authors expect that a shift to glucose might show a stronger effect on GID/Ubc8 and DHFR import?

What is the evidence for a significant import defect in the absence of Ubc8, rather than an (additional) effect on gene expression (transcription and/or translation) of mitochondrial proteins? This point may be supported by the fact that mitochondrial proteins are decreased in lactate grown *ubc8* cells in figure S2D, a condition with presumably low Ubc8/GID complex activity.

To distinguish between these different factors, the authors should determine from their SILAC proteomics data protein synthesis and turnover rates for mitochondrial proteins in WT vs *ubc8* cells in lactate or lactate + glucose conditions. Importantly, do the authors detect an accumulation of precursor proteins (corresponding peptides) in their proteomics analysis of *ubc8* cells compared with WT? Could these data link accumulation of precursor protein to reduced steady state levels of mitochondrial proteins?

Figure 2 A-D: it is not indicated whether light or heavy or both peptides/proteins were analyzed or are shown. It would be interesting to show representative examples for mitochondrial proteins as in Fig. 2E for old and new proteins for WT vs *ubc8*.
Figure S2A: there are no protein names indicated.

Figure 3A-D should probably go to supplementary, as it confirms published data.

Figure 4A: Western blots need to be quantified and normalized for loading to be convincing.

Does mitochondrial morphology change upon diauxic shift? The authors tested long-term adapted cells in galactose medium. However, *Ubc8* seems to play a role in the transition of metabolic states of cells.

We would like to thank all three reviewers for the thorough evaluation of our manuscript. We were very pleased to see that all three referees in general were positive and raised a number of critical points which we addressed by additional data (Fig. 1G, 4E, 4F, 5B, 5C, S2A, S2B, S3C, S4B, S5C, S5D, S6A-E, S7A-D) and by changes of the text.

In particular, we followed the suggestion of all three referees to better characterize the metabolic conditions under which Ubc8 is relevant for Tom22 biogenesis and mitochondrial biogenesis. To this end, we performed another complex proteomics analysis of wild type and $\Delta ubc8$ cells continuously grown in glucose (this was particularly requested by referee #1), galactose and lactate, respectively. This showed that Ubc8 is particularly important upon changing metabolic conditions, thus when non-fermentable carbon sources are replaced by glucose-containing medium. Furthermore, we tested whether mitochondrial precursor proteins can be detected in cell extracts by Western blotting. We indeed observe these precursor forms at levels that are comparable to those in mutants lacking Ubx2 (Martensson et al. 2019. Nature 569, 679f). The results confirm the reduced import efficiency in $\Delta ubc8$ cells, again supporting our conclusion.

With these additional data and by many text changes, we hope that we were able to satisfactorily address all points raised by the referees.

Point-by-point response to the comments of the referees

Reviewer #1:

We thank the referee for her/his very positive evaluation and for the comment that 'overall, the paper is well written and the data confirming the previously described role of Ubc8 in catabolite degradation upon transition to fermentative growth as well as its new function in Tom22 protein biogenesis are convincing.' We addressed the many really helpful and specific comments as described in the following:

Major comments:

1. The authors show that deletion of Ubc8 increases the cytosolic levels of two artificial mitochondrial fusion proteins: Oxa1-Ura3, which was used to identify yeast mutants (including Ubc8) that slow mitochondrial protein import (this study and Hansen et al., 2018), and b2(1-167) Δ 19-DHFR, a fusion construct widely used to assess mitochondrial protein import in vitro. Based on these results, the authors conclude that loss of Ubc8 causes "increased amounts of mitochondrial precursor proteins in the cytosol". As it stands, the data presented in the manuscript do not sufficiently support this conclusion:

First, while b2(1-167) Δ 19-DHFR levels are indeed elevated in $\Delta ubc8$ cells it remains unclear if this pool is cytosolic or mitochondrial. Moreover, whether the observed delay in b2(1-167) Δ 19-DHFR turnover in $\Delta ubc8$ cells is due to impaired degradation in the cytosol or within the mitochondrial matrix is not assessed. Of note, the immunoblot (Figure 1D-E) at best shows a

minimal reduction in b2(1-167) Δ 19-DHFR turnover in absence of Ubc8 and the quantification only significantly differs at a single timepoint.

Second, whether deletion of Ubc8 indeed causes global retention of mitochondrial proteins within the cytosol is not assessed. This is surprising, as the authors subsequently use proteome analysis to analyze differences in the cellular proteome of wild-type (WT) and Δ ubc8 cells upon shift from lactate to glucose. The authors should at the very minimum assess the accumulation of mitochondrial protein precursors in the cytosol using subcellular fractionation and/or Western Blot analysis. Unprocessed mitochondrial preproteins should be readily detectable by Western Blot

Third, in Figures 2E and 3A-D the authors demonstrate Ubc8-dependent degradation of gluconeogenic enzymes upon switch to fermentative growth. Surprisingly, the effect of Ubc8 on mitochondrial protein turnover is not further elucidated. Therefore, whether Ubc8 "promotes the biogenesis of mitochondrial proteins" as claimed or has a (potentially additional) degradation-based effect remains unclear. Since the data is already available, the authors should include this analysis in an updated version of the manuscript and should also assess whether any observed decline in mitochondrial protein levels is sensitive to cytosolic proteasome inhibition.

Following the suggestion of the referee, we performed Western blots on cellular extracts of wild type and Δ ubc8 cells using antibodies against Mdj1 and Rip1, two proteins for which cytosolic precursors could be observed in the past (Mårtensson et al. 2019 Nature 569, 679f; Sahi et al. 2013 Mol Biol Evol 30, 985f). As shown in the novel Figs. 1G and S2A, we indeed observed increased amounts of Mdj1 and Rip1 precursors in the Δ ubc8 mutant, comparable to the precursor levels in Δ ubx2 mutants or upon clogger expression, two controls for which such precursors were reported before. We thank the referee for this suggestion.

However, in our hands the amounts of these precursors are always rather low, in particular in comparison to the amounts of mature proteins. We therefore removed the statement about accumulating precursor proteins from the abstract.

2. The authors convincingly show that deletion of Ubc8 impairs cell survival in glucose-grown cultures under stationary conditions (Figures 3H and S3B), impairs metabolic adaptation upon respiratory-to-fermentative transition, and demonstrate an overall reduction in mitochondrial protein levels (Figure 2D) in Δ ubc8 cells. Based on these results, the authors infer a "reduced functionality of mitochondria in Δ ubc8 cells". However, a clear link between mitochondrial dysfunction in Δ ubc8 cells and the observed metabolic defects remains missing. Notably, Δ ubc8 cells exhibit no growth defects on respiratory carbon sources (Figure S1E) suggesting that mitochondria are indeed fully functional in absence of Ubc8. This latter discrepancy may be partially explained by the fact that the effect of Ubc8 on Tom22 levels "was less pronounced in glycerol grown cells". Unfortunately, biochemical data supporting this conclusion is not provided in the manuscript. This discrepancy needs to be addressed.

We agree with the referee that mitochondria in $\Delta ubc8$ cells are functional and able to respire. Thus, Ubc8 is not essential for respiration. We actually explicitly write this in the text. However, as shown in our study, in the absence of Ubc8 the levels of Tom22 are reduced and therefore mitochondrial biogenesis is not as efficient as in the presence of Ubc8. We agree with the referee that Ubc8 is not essential for respiration nor for import. However, the CCCP sensitivity experiment shown in Fig. 1F shows that $\Delta ubc8$ cells show an increased sensitivity against uncoupling of the mitochondrial inner membrane, indeed indicating that the respiration machinery is less active in proton pumping than that of wild type cells.

Along those lines, the authors should test if temperature sensitive alleles of Tom22 phenocopy the observed metabolic effects of $\Delta ubc8$. This would allow the authors to uncouple the mitochondrial function of Ubc8 (Tom22 biogenesis) from its role in the GID complex.

We followed the suggestion of the referee and tested whether a temperature-sensitive *tom22* mutant shows similar phenotypes as $\Delta ubc8$ cells in respect to their CCCP sensitivity and their death rates in stationary phase. This is indeed the case, and the temperature-sensitive *tom22-102* mutant apparently phenocopies what we observed with $\Delta ubc8$ cells. We show the results here for inspection by the referee.

Fig. 1 for referee #1: **A.** WT and *tom22-102* cells were precultured in galactose medium to mid-log phase. Tenfold serial dilutions were dropped onto plates containing glucose, galactose or glycerol as carbon source. The plates were incubated at 25°C, 30°C and 37°C. **B.** WT and *tom22-102* cells were spread on glucose plates and a filter was placed to the center to which 10 µl 10 mM CCCP was added. Cells were incubated for two days, and the inhibition area (A) was measured. **C.** Cells were grown in 2% glucose medium to full saturation (about OD 5) and further incubated in a shaker at 30°C for 10 days. At day 1, 4, 7 and 10, an aliquot of the culture was analyzed and viable cells counted after plating on glucose medium. Shown are mean values of three biological replicates.

3. The authors demonstrate that Ubc8 is required for maintaining mitochondrial Tom22 levels and that deletion of Tom22 impairs its incorporation into the TOM complex. However, whether this block occurs at the step of complex assembly (as concluded by the authors) or earlier (i.e. at insertion into the outer membrane) remains unclear and should be addressed. Of note, the levels of the SAM-Mdm10 complex which mediates Tom22 assembly are reduced in Figure 5A, but the authors state that SAM levels are unaltered in Δ ubc8 cells. This should be changed. Along those lines, it seems surprising that despite the strong reduction in Tom22 and other mitochondrial protein levels (noted in Figure 2D), as well as the claimed import defect caused by the lack of Tom22, all other mitochondrial proteins shown in the blot appear to be unaffected by Ubc8 knockout. Please add text to address this discrepancy.

In order to better resolve whether insertion or assembly of Tom22 is impaired in the Δ ubc8 mutant, we performed import experiments with a radiolabeled Tom22 version in which the C-terminus is extended by three methionine residues. Proteinase treatment of this variant produces a characteristic fragment (Tom22') which allows it to measure the insertion of Tom22 into the outer membrane independent of its subsequent assembly into the TOM complex (Ellenrieder et al. 2019 Mol Cell 73, 1056f). As shown in the novel Fig. 5B and C, the insertion of Tom22 into the outer membrane of Δ ubc8 mutant mitochondria is indistinguishable from that in wild type organelles. We therefore conclude that it is the assembly step that is affected if Ubc8 is absent and not the outer membrane insertion of the Tom22 precursor per se. We thank the referee for pointing out to us the change in the SAM-Mdm10 complex in Figure 5A and now mention this in the text.

4. In the discussion the authors speculated that Ubc8 may mediate its effect on Tom22 levels through degradation of a cytosolic phosphatase or kinase that antagonizes Tom22 import. However, as Figure 5B shows reduced Tom22 assembly into TOM in vitro using radiolabelled Tom22 and isolated mitochondria without the presence of cytosolic enzymes, this seems counterintuitive. Please add supporting data or adjust the manuscript text accordingly

We rewrote the respective parts of the discussion and fully agree with the referee, that on basis of our observation an indirect effect via changes in the phosphorylation state of the cytosolic Tom22 precursor seems unlikely.

5. If possible, please include a comparison of the proteome of wildtype and Δ ubc8 cells continuously grown in glucose.

We performed this comparison as well as measurements of cells continuously grown on galactose and lactate. These novel datasets are now shown as novel Figs. S7A-D. These data indicate that Ubc8 has only minor relevance in continuously grown cultures, regardless of whether cells grow under fermentative or respiratory conditions, however, Ubc8 is critical when cells switch from respiration to fermentation. Thus, the Ubc8-mediated stimulation of the TOM assembly is, just like the degradation of glycolytic enzymes, specifically critical when the metabolic conditions are altered.

Minor comments:

In Figure S3A the authors assess MGO levels after 1h of growth on glucose. However, growth differences between wildtype and Δ ubc8 only become apparent after 20-48h. Do MGO levels remain unchanged in Δ ubc8 cells at later timepoints? This analysis is critical to conclude that toxic metabolites do not contribute to increased lethality in the stationary phase. In case it is not feasible to include these data, please adjust the conclusions accordingly.

We now added an additional figure panel (Fig. S4B) for which we measured the methylglyoxal levels 24 h after the metabolic shift as suggested. Again, there were no significant differences between wild type and Δ ubc8 cells observed.

The authors assess import of Pet9 by SDS-PAGE and conclude its biogenesis is impaired in Δ ubc8 cells (Figure 4A). However, upon import mitochondrial carriers assemble into multi-subunit complexes and thus, their biogenesis is best assessed by BN-PAGE. Please include this analysis.

The loss of Ubc8 leads to a depletion of Tom22 and changes in the composition of the TOM complex. This is why we specifically tested whether the translocation of carriers across the TOM complex is altered (rather than their biogenesis in general). Therefore, we followed the protease-resistance after protein import, what is also referred to the transition from stage II to stage III in the carrier import field (e.g. Kübrich et al. 1998 JBC 273, 16374f).

The increase in the Oxa1 precursor form (Figure 4A) points to additional import defects after the precursor has been translocated across the outer membrane. Please address this in the manuscript text.

In this experiment there were indeed very minor amounts of Oxa1 precursor seen after protease treatment of Δ ubc8 mitochondria. However, we do not think this is a consistent

phenomenon. We repeated this type of import experiment many times. Whereas the reduced import rates were seen repeatedly in many repeats with different mitochondrial preparations, the amounts of precursor background after protease treatment varied. Therefore, we decided not to mention this explicitly.

The manuscript states that Δ ubc8 cell growth rate lags behind following (but not before) the diauxic shift. It would be helpful to include language or data showing how the timepoint of the diauxic shift was determined.

The diauxic shift is defined as a reduced growth rate during log phase leading to a bi-phasic growth behavior. We now described this in the text and added arrows to indicate the shifts better in the figure.

For several figures, most prominently the immunoblots, data on the number of repetitions should be included.

We included these numbers as requested for all experiments in which data were quantified.

Several Figures and Figure Legends would benefit from additional or improved labelling:

1. Figure 1C is lacking a figure legend to distinguish the empty vector controls

We added these.

2. Please annotate the other well-growing strain shown on the plate in Fig S1B (likely Ema35?) and mention it in the manuscript text.

We annotated it now. It is mentioned in the figure legend.

3. Figure S1D is not addressed or referenced in the text at all.

We included the reference.

4. Information about the employed statistical tests and significance levels is missing in Figure 1E, as well as in the proteomics data (Figures 2C, 4B, S2A and S4B). Please clarify which data points reflect significant outliers.

We added information about the statistical analysis (Fig.1E) in the figure legend. We also indicated the trend lines for significance levels to Figs 2C. However, in particular in the complex dynamic SILAC proteomics data, these levels depend on the way the data are normalized. Therefore, we do not refer to lists of significantly changed proteins but

always show the entire proteome scatter for which the position of individual proteins can be assessed in comparison to all other proteins.

5. In Figure 2C, the annotation of the gluconeogenic proteins is difficult to see and the annotation of Jen1 in the first plot is not explained and missing in the second plot.

We removed the labeling for Jen1.

6. Figure S2A is lacking the names of mitochondrial proteins referenced in the figure description.

We changed the description accordingly.

7. Figure 2D: Please consider moving Figure S2B to the main Figure as the data is very convincing.

We moved this Figure panel as suggested and thank for the positive comment.

8. In Figure 2D, please mark mitochondrial proteins with a line next to the plot instead of referring to a vaguely to "many" mitochondrial proteins. Consider separating these out in an independent heat map.

Positions of mitochondrial proteins are indicated already in the figure on the left. We now wrote 'Enrichment of mitochondrial proteins' as 'Many mitochondrial proteins' was indeed somewhat sloppy. We also included a novel Figure showing the intensities of individual mitochondrial proteins to make the relevance of Ubc8 for these proteins better visible (novel Fig. S3C).

9. In Figure 2D it is not clear whether "protein levels increased/reduced in Δ ubc8" refers to the t0 lactate state, the t1 shifted cultures or both.

We now added 'in comparison to wild type cells' to make this sentence clearer.

10. In Figure 2F, comparisons of mitochondrial proteins with other compartments such as ER or peroxisomes would be helpful to assess whether Ubc8 selectively affects the mitochondrial proteome.

Proteins of the ER or the endomembrane system were not enriched. However, those of the peroxisome were also enriched, similarly to those of mitochondria. We added this information to the figure (which is now Fig. 2G in the revised version).

11. Please explain what the additional Icl1 band is that appears in Δ ubc8 cells (Figure 3C).

We do not know why some of the Icl1 protein shifts in size. However, we observed this in many experiments very consistently. From the size, it would be comparable with a modification by one ubiquitin or SUMO protein. Upon long exposure, we saw similar adducts also for Fbp1 and Mdh2. Since we did not want to speculate about the nature of these forms, we did not discuss these size shifts in the text. However, we also did not cut these higher forms off so that careful readers can see these modifications.

12. It would be helpful to include numerical/statistical data on the exact timepoint of when WT and Δ ubc8 cells enter oscillation and the cycle time should be included in the manuscript and/or the figure (Figure 3G)

As shown in the figures, the cycle times are not constant but get shorter over time. This is why we decided to show the entire runs from two independent measurements because we felt that thereby the effects of the Ubc8 deletion are clearer than if we would have shown bar graphs.

13. The figure description of Figure 3H describes that "cultures were grown to mid-log phase in glucose medium" while the text refers to a switch "from glycerol medium", this should be corrected.

We corrected this. We thank the referee for the very careful proof-reading!

Reviewer #2:

We thank the referee for the very positive evaluation and for her/his statement that 'the presented experiments are of a high technical quality and acceptance of this manuscript is recommended'. We addressed her/his comments as follows:

Figure 1C would benefit from a legend indicating, which color corresponds to what strain. Since the 'WT' label is currently in between three lines in the graph it is slightly ambiguous. Upon comparing Fig S1A and S1C, it seems as if the WT+ Δ N-Oxa1-Ura3 strain grows on medium lacking Uracil on the drop dilutions (Fig. S1A), but not in the growth curve (Fig S1C). However, this could be due to my misinterpretation of the colors in Figure S1C, which could be changed to make them easier to distinguish.

We added a legend as suggested.

While evident for an expert reader, the purpose of the Δ N-Oxa1-Ura3 construct could be briefly explained.

We added an explaining sentence as suggested.

The authors should discuss the data presented in Figure 1E in more detail. Why is there an effect on Su9-DHFR in the first 30 minutes, but not after that in Δ ubc8 cells? Removal of the remaining precursor even seems to speed up after the initial 30 min.

We added some more explaining comments. As indicated by the error bars in the quantification we saw some fluctuations in different experiments. The levels of the b₂(1-167) Δ ₁₉-DHFR precursors were consistently higher in Δ ubc8 cells than in wild type cells. However, only for the 30 min time point this difference passed the significance test.

The experiment presented in Figure 1F should be performed in triplicates.

We performed this experiments several times and decided to show one representative experiment.

The oscillatory oxygen consumption (Figure 3F) should be explained in slightly more detail, which would be helpful to the reader. How can the oscillation of oxygen consumption be explained (even for WT)? Along the same lines, it is not clear to me why the shorter cycles in Δ ubc8 cells are 'consistent with a disrupted ability to switch between different metabolic states'.

We describe the experiment now in much more detail and included a number of references in the text.

The text should briefly explain the potential toxic effects and origin of methylglyoxal.

We added explanatory text and a reference.

Ubc8 is intimately linked to adaptation of cells to changing metabolic requirements. The authors of the present study confirm this by demonstrating that loss of Ubc8 leads to impaired removal of Gluconeogenesis enzymes upon shift from respiratory to fermentative growth. The authors show clearly that deletion of UBC8 affects accumulation of Tom22 and compromises mitochondrial import. Whether these effects can also be linked to a metabolic shift and whether this is a conclusion that the authors draw, however, remains less clear to me at this stage. The screen presented in Figure 1 was performed on Glucose, I assume, and thus deletion of UBC8 leads to defective import and accumulation of Oxa1-Ura3 on a fermentable carbon source. Clearance of Su9-DHFR precursor is compromised during a shift from Galactose to Glucose in Figure 1D. Tom22 appears decreased in Δ ubc8 cells in cells grown on glycerol (Figure S4B) and upon shift from Glycerol to Glucose medium (Figure 4B). Thus an effect on mitochondrial

protein import is somewhat apparent under all tested conditions. The authors should try to clarify if Ubc8 is important for mitochondrial import and TOM complex assembly generally or only upon metabolic shifts. In particular, it would be important to assess Tom22 protein levels in WT and Δ ubc8 strains by Western Blot in cells continuously grown on YPG. Is the result different from the data presented in Figure 4C? In the current form of the manuscript, it is difficult to tell whether there is a difference regarding Tom22 levels between cells grown in Glycerol or cells shifted from Glycerol to Glucose by comparing figures 4B and S4B. Alternatively, Ubc8-dependent Tom22 regulation upon a metabolic shift could be determined by collecting multiple samples over the course of a shift from glycerol to glucose in WT and Δ ubc8 cells. Similarly, in organello import experiments could be performed with mitochondria isolated from glycerol or after a shift onto glucose. Along these lines, it should be stated for Figure 4A from which growth medium mitochondria were isolated to perform the import.

In order to better characterize whether Ubc8 is relevant on continuously grown cultures, we performed an additional complex mass spec experiment which now is shown as novel Fig. S7A-D. We grew wild type and Δ ubc8 cells to log phase in three different conditions: continuously in 2% glucose, continuously in 2% galactose and continuously in 2% lactate. From four independent samples each, whole cell extracts were prepared and analyzed by label free mass spectrometry. At all conditions, the levels of Tom22 were reduced in the absence of Δ ubc8. However, the reduction was considerably smaller than in the proteomes of cells that had been changed from lactate to glucose (Fig. 4B-D, Fig. 5D, E). Thus, in respect to its role in Tom22 assembly, Ubc8 seems to be particularly relevant upon changing conditions. This is consistent to the activity of Ubc8 in degradation of gluconeogenesis enzymes which is also strongly increased upon metabolic changes.

As requested, we analyzed Tom22 protein levels in wild type and Δ ubc8 cells grown in glycerol medium by Western Blot (Fig. S5C, D). Tom22 levels were reduced in Δ ubc8 cells as well. Thus, both in whole cells and in isolated mitochondria, reduced Tom22 levels were found when Ubc8 is absent.

We added information about the metabolic conditions at which the cells were harvested for the different experiments throughout the study.

A reduction of Tom22 protein levels in Δ ubc8 cells could have several reasons. While Figure 5B suggests that Tom22 accumulation is caused by compromised assembly into the TOM complex, a potential transcriptional response in the absence of Ubc8, should be excluded by performing RT-qPCR experiments to assess the Tom22 mRNA levels in WT and Δ ubc8 cells grown in glycerol or upon a shift from glycerol to glucose.

Following the suggestion of the referee, we added the qPCR experiment and show the results in the novel Fig. 4E and F. The levels of *TOM22* mRNA are not altered by Ubc8 deletion.

A brief discussion about the change in the TOM complex size in $\Delta ubc8$ (Figure 5A) would be helpful. What change could lead to this size shift? Figure 5C suggests that it could be a conversion of the dimeric into a monomeric version of the TOM complex, this however seems unlikely due to the rather small shift.

We agree with the referee that a conversion of a dimer to monomer is unlikely. We therefore changed the model accordingly and improved the discussion in the text. Since the size shifts are minor (though highly consistent) we assume that one of the Tom22 subunits and potentially some of the small Tom subunits are lacking in these faster migrating isoforms.

Figure 5B shows that in organello assembly of radiolabeled Tom22 into the TOM complex is compromised. In this experiment, cytosolic components should be absent from the reaction. Therefore, it seems unlikely that cytosolic kinases or phosphatases play a direct role on Tom22 for its assembly. This point should be discussed.

We changed the discussion in the text and the final model accordingly. See also our response to point number 4 of referee #1.

The text refers to Figure S4C by stating that absences of Ubc8 leads to ' wide-ranging depletion of mitochondrial proteins, in particular those of the matrix for which the Tom22 levels seem particularly important.'. It may be the case that I am not an expert in the interpretation of violin-plots, but to me it does not appear clear where in the plot I would find support for this statement. Perhaps the figure could be improved by highlighting the parts of the plot that support this conclusion. Further information could be given in the legend.

We added now percentage scores into the figure to make it clearer. With this comment we wanted to stress that many matrix proteins are reduced in $\Delta ubc8$ cells, whereas proteins of the outer membrane and the intermembrane space are not affected or even relatively increased. However, the degree of reduction or increase is not strong. This is why we wrote that the depletion is only moderate, even if many proteins are affected (hence it is wide-ranging). In order to avoid the impression that the data are overinterpreted, we now deleted the word 'wide-ranging'. We hope that with the percentage scores this supplemental figure is now clearer.

Minor points:

The authors should indicate the precursor and mature forms of DHFR in Figure 1D.

We changed the figure.

Figure 3H legend - "cultures were grown to mid-log phase in glucose medium..." is inconsistent with the main text on page 9 ("Next we tested the growth of cells after switching from glycerol medium to different concentrations of glucose"). This should be rectified by the authors.

We corrected the text. See also question 13 of referee #1.

Typo: Reference to Figure 3G in the text says 'read arrows' instead of 'red arrows'. It should be added to the legend of the figure, what the red arrows exactly point at.

We corrected the text and now wrote: 'The red arrows point to the phase of adaptation between starvation and induction of oxygen consumption that initiates metabolic cycling.'

Legend Figure S2 states that 'Names of significantly changed mitochondrial proteins are indicated'. This is not the case and should be corrected.

We corrected the text.

No real explanation is given for Figure S1D. What is the relevance of the comparison of Ubc8 to other proteins?

We show the overall organization of Ubc8 in this figure panel. This might not be novel for readers of the ubiquitin ligase field. But since we expect that many readers will have a mitochondria background, we decided to keep this figure included as supplemental figure item.

Reviewer #3:

We thank the reviewer for the overall positive assessment. We addressed the specific points in the following way:

The authors use a number of different growth conditions, which make it difficult/confusing for the reader to compare results/experiments with each other. Conditions are not always indicated in the figure legends, and it is often not explained why the authors use a particular condition for certain experiments.

Since the same comment was also made by referee #2 (see above) we performed an additional dataset for which we measured the proteomes of wild type and $\Delta ubc8$ cells after growth on different carbon sources (glucose, galactose, lactate). These additional

data (Fig. S7A-D) show that Ubc8 is less important when cells are consistently grown, regardless of whether they grow by fermentation or by respiration.

In addition, we improved the description of the experiment conditions in the figure legends of the revised version.

The authors propose that Ubc8 is a positive regulator of Tom22 assembly in TOM complex and mitochondrial import. However, the Ubc8/GID complex is activated when glucose is present, which represses mitochondrial biogenesis. How do the authors explain this counterintuitive connection? Is there a particular subset of mitochondrial proteins that is affected by the potential Ubc8-Tom22 regulatory arm, which could aid in mitochondrial adaptation to altered carbon/growth conditions?

The switch from respiration to glucose leads to a considerable increase of cell division (1.5 h vs. 4.5 h) and a considerable increase in protein synthesis rates. During transition, when mRNAs for OXPHOS proteins are still present at high levels, the increase in the translation rate might lead to a temporary accumulation of precursors. We assume that the Ubc8-mediated induction of TOM assembly prevents the overload of the import system during this transition. Since this is just a hypothesis and we do not have data on the mRNA and precursor abundance under different growth conditions, we avoided a speculation about this aspect in the discussion of our study. However, we intend to address this interesting aspect further experimentally as well as by modeling on basis of available quantitative data in the future.

The authors describe a novel and interesting link between the GID complex and TOM complex assembly with clear effects on Tom22. However, the molecular underpinning of this connection remains elusive. Specifically, it is unclear whether the GID complex directly affects Tom22 or other regulators of Tom22/TOM assembly. Given their expertise in proteomics, would the authors be able to define ubiquitination patterns by reanalyzing their whole cell proteomics data to determine potential effects of Ubc8 on candidate proteins?

In our proteomics datasets we only found about 70 peptides with GG signatures indicating previous ubiquitination. The low coverage did not allow us to identify potential GID substrates. For identification of such peptides, proteasomal degradation has to be blocked and ubiquitinated peptides have to be enriched by affinity purification. Given that we only had three months for the revision, such a complex experiment was not possible. However, we agree with the referee that this is now the next important step to do and we will follow this up in the future.

Given that the molecular mechanisms remain elusive, the term "regulate" in the title seems to be a bit of an overstatement at this point. I am aware that "regulate" is often (over)used instead of the more appropriate "affect". To me, however, "regulate" implies a direct regulatory role, for

example, direct Ubc8-dependent ubiquitination of Tom22 or its assembly factors. This, however, has not been demonstrated, yet.

We agree with the referee that the term regulation might imply that Ubc8 adapts Tom22 levels to specific conditions for which we have no evidence. We therefore changed the title and replaced 'regulates' by 'promotes', which is exactly what we show here.

Along these lines, does Ubc8 overexpression increase Tom22 assembly into the TOM complex and enhance protein import into mitochondria?

We followed the suggestion of the referee and tested the consequences of Ubc8 overexpression. As shown in the novel Fig. S7A-E, overexpression of Ubc8 leads to a growth defect on glycerol, lower levels of Tom22 and import-deficient mitochondria. This supports the functional connection of GID complex activity and the structure and function of the TOM complex. However, in these experiments, the levels of Ubc8 are unphysiologically high, certainly much higher than upon physiological conditions.

The effect of *ubc8* on Tom22 and TOM complex assembly could explain the observed changes on mitochondrial proteins. However, the authors do not provide experimental evidence whether this is sufficient to explain the defects observed in the absence of Ubc8. For example, does TOM22 overexpression (at least partially) rescue (some of) the *ubc8*-associated phenotypes?

Unfortunately, we did not obtain convincing results about a potential rescue of the *ubc8* mutant by Tom22 overexpression due to technical problems. We would have needed more time than the three months granted by the journal. However, following the suggestion of the referee to look for a genetic interaction of Ubc8 and TOM subunits, we realized that Ubc8 could not be deleted in mutants lacking the TOM subunits Tom6 or Tom7. This suggests that double deletion mutants are inviable. However, since we had no time to control this (e.g. by tetrad dissection) and to characterize this genetic interaction, we decided not to include these data into the manuscript but to show here for inspection by the referee:

G418 plates of *UBC8* knockout in:

Fig. 2 for referee #3:

The *UBC8* gene was deleted by a kanamycin resistance cassette. Whereas this approach yielded in viable deletion mutants with wild type cells, *UBC8* deletion mutants were not obtained in $\Delta tom6$ and $\Delta tom7$ mutants.

Based on figures 1D and E, the authors conclude that the DHFR-reporter is imported significantly more slowly in *ubc8* mutants compared with WT cells. However, only one data point (30 min) shows statistical significance. Thus, the effects on mitochondrial import are rather minor. Why do the authors shift cells from galactose to lactate? Wouldn't the authors expect that a shift to glucose might show a stronger effect on GID/Ubc8 and DHFR import? What is the evidence for a significant import defect in the absence of Ubc8, rather than an (additional) effect on gene expression (transcription and/or translation) of mitochondrial proteins? This point may be supported by the fact that mitochondrial proteins are decreased in lactate grown *ubc8* cells in figure S2D, a condition with presumably low Ubc8/GID complex activity.

To distinguish between these different factors, the authors should determine from their SILAC proteomics data protein synthesis and turnover rates for mitochondrial proteins in WT vs *ubc8* cells in lactate or lactate + glucose conditions. Importantly, do the authors detect an accumulation of precursor proteins (corresponding peptides) in their proteomics analysis of *ubc8* cells compared with WT? Could these data link accumulation of precursor protein to reduced steady state levels of mitochondrial proteins?

Following the suggestion of the referee, we now added a figure in which the expression levels of a number of mitochondrial proteins was shown that we deduced from the SILAC experiment as suggested (novel Fig. S3C). It is evident from the figure that for the proteins shown the levels were higher in wild type than in $\Delta ubc8$ cells. In most cases, the shift to glucose reduced the levels of these proteins, presumably due to the glucose-repression of these proteins. However, even upon constant lactate conditions, the levels of these proteins were reduced if Ubc8 was deleted. This is consistent with the reduced import efficiency and the defects in TOM assembly of $\Delta ubc8$ mitochondria. Moreover, we tested whether precursor proteins can be detected in $\Delta ubc8$ cells by Western blotting. Please see our comments to point 1 of referee #1 to this aspect. Such accumulating precursors are indeed observed in $\Delta ubc8$ cells, albeit at low levels. These data are now shown in the novel Figs. 1G and S2A. We also change the text and

discussion accordingly. In order to avoid an overinterpretation of the data, we removed the statement about accumulating precursors from the abstract.

Figure 2 A-D: it is not indicated whether light or heavy or both peptides/proteins were analyzed or are shown. It would be interesting to show representative examples for mitochondrial proteins as in Fig. 2E for old and new proteins for WT vs *ubc8*.

We show this now as Fig. S3C.

Figure S2A: there are no protein names indicated.

We corrected the text. See also question 13 of referee #1.

Figure 3A-D should probably go to supplementary, as it confirms published data.

We decided to show these blots as they make it easier to understand our story even though the main message here confirms previous results.

Figure 4A: Western blots need to be quantified and normalized for loading to be convincing.

This figure shows an autoradiography, not a Western blot. For comparison, lanes with 20% of the radiolabeled precursor used per experiment are shown in the first lane. For all Western blots used in the study, loading controls are shown.

Does mitochondrial morphology change upon diauxic shift? The authors tested long-term adapted cells in galactose medium. However, *Ubc8* seems to play a role in the transition of metabolic states of cells.

We did not find any relevance of *Ubc8* for mitochondrial morphology. During diauxic shift, the mitochondrial network expands but the overall mitochondrial morphology is not changed. This was true both in wild type and in Δ *ubc8* cells. We include a figure showing the mitochondrial network in wild type and *Dubc8* cells during diauxic shift (as controlled by detection of growth rates in a plate reader) for inspection of the referee.

Fig. 3 for referee #3:

WT and $\Delta ubc8$ cells expressing a mitochondria-targeted GFP protein were grown in glucose medium in a plate reader until stationary phase. Cells were harvested and mitochondrial morphology was visualized by fluorescence microscopy.

September 26, 2022

RE: Life Science Alliance Manuscript #LSA-2022-01526-TR

Dr. Johannes M Herrmann
University of Kaiserslautern
Cell Biology
Erwin-Schroedinger-Strasse 13
Kaiserslautern D-67663
Germany

Dear Dr. Herrmann,

Thank you for submitting your revised manuscript entitled "The metabolite-controlled ubiquitin conjugase Ubc8 promotes mitochondrial protein import". We would be happy to publish your paper in Life Science Alliance pending final revisions necessary to meet our formatting guidelines. Please revise and format the manuscript and upload materials by Thursday.

- please address the remaining comments from Reviewers 1 and 2
- please upload your graphical abstract as a separate file and label it as graphical abstract
- please double-check your figure callouts for Figure S1; you have a callout for panel F, but this is not in the legend and seems like it should be panel E
- please add a callout for Figure 5C and Figure S3C to your main manuscript text
- please add your supplementary figure legends and table legends to the main manuscript text
- please remove the Reviewer account details section and make the uploaded datasets publicly accessible

A. FINAL FILES:

B. MANUSCRIPT ORGANIZATION AND FORMATTING:

Thank you for your attention to these final processing requirements. Please revise and format the manuscript and upload materials by Thursday.

Sincerely,

Reviewer #1 (Comments to the Authors (Required)):

In their revised manuscript, Roedel et al have done a very good job in addressing my previous concerns and that of the other referees. I recommend publication of the revised manuscript after the following remaining minor issues have been satisfactorily addressed:

Page 6. Figure 1G is incorrectly referenced as 1F, and vice versa

Page 6. Why do the authors cite Figure S2A with Figure S2B? It appears that Figure S2A shows an orthogonal approach to Figure 1F, while Figure S2B demonstrates the lack of RPN4 response. It would therefore seem more intuitive for Figure S2A to be cited alongside Figure 1F in the text.

Page 7. Figure S1E is incorrectly referenced as S1F

Page 8. Please highlight Fbp1, Mdh2, Icl1, and Pck1 in Figure S3A, otherwise their protein levels can not be compared between Fig. 2C and Fig. S3A without inspecting the source data.

Page 12. The authors should add an explanation why overexpression of Ubc8 phenocopies the effects of Ubc8 deletion. As it stands, the data is counterintuitive.

Page 13. Please clarify „hours" in the sentence "cells that were shifted hours from glycerol (respiration) to glucose (fermentation)"

Page 13. The authors should add a brief sentence regarding the role of the SAM-MDM10 complex in Tom22 insertion into the TOM complex at the end of the second paragraph. Reduced SAM-MDM10 levels support the authors conclusion that ubc8 promotes Tom22 assembly but not import.

Page 14. Fig. 5B should be Fig. 5B+C

Page 14. Please add a label explaining what the percentages in the violin plots shown in Figure S5C refer to. It is explained in the figure legend, but readers would likely benefit from additional labelling in the figure itself.

Page 16. The authors speculate about a possible role of Ubc8 in regulation of Por1 (which itself regulates Tom22 biogenesis). Do the authors find any support for their hypothesis in the MS data. If so, this should be included in the discussion.

Figure 1G. The authors should add a brief explanation of how deletion of Ubx2 is expected to affect mitochondrial precursor

levels

Figure 2. Fig S3A is incorrectly cited as S2A in the Figure 2 figure legend

Figure 4. Figure 4B is incorrectly labelled as 4E in the figure legend

Fig 5B/D. Please clarify the asterisk (*) in the figure legend

Figure S4A/B. The authors should add a brief explanation for the Glo1 deletion mutant. Is it expected that the wt and Glo1 deletion strains have similar levels of MG at 24h?

Fig S5D is incorrectly labelled as E in the figure legend („Panel E shows...“)

Figure S7 is only referenced in the discussion. Please reference it in the results section when comparing proteins levels after metabolic shift to steady state levels.

Table S4. The table is not referenced in the text and provides source data for Fig.S6B-D. Presumably the authors mean S7B-D?

Reviewer #2 (Comments to the Authors (Required)):

As stated in my first statement, the submitted manuscript presents novel findings on the role of Ubc8 in facilitating metabolic adaptation of yeast cells. Therefore, I support publication of this manuscript after considering the points below.

As a note at the beginning, I was not able to find the figure legends for the supplementary material and can therefore not make any statements about the accuracy and quality of this part.

The authors satisfactorily address many of the comments raised by reviewers. Several parts in the text were edited, which now allows the reader more easily to understand under which conditions Ubc8 affects protein import.

In general, it is evident in the presented work that presence of Ubc8 is required for wild-type-like mitochondrial import and in particular for the assembly of the TOM complex and accumulation of wild-type-like Tom22 levels. How Ubc8 acts to support these processes remains currently unclear and addressing this question is outside of the scope of this work. In this light, however, I would like to second the following remark made by reviewer #3:

"Given that the molecular mechanisms remain elusive, the term "regulate" in the title seems to be a bit of an overstatement at this point. I am aware that "regulate" is often (over)used instead of the more appropriate "affect". To me, however, "regulate" implies a direct regulatory role, for example, direct Ubc8-dependent ubiquitination of Tom22 or its assembly factors. This, however, has not been demonstrated, yet."

It could well be that absence of Ubc8 affects assembly of the TOM complex by indirect means. Therefore, I find the statements at the bottom of page 12 ("Thus, the GID complex....controls") and at the bottom of page 14 ("Thus, ... the TOM complex, is under the control of Ubc8") slightly misleading as it implies a direct role of Ubc8. Furthermore, I agree with reviewer #3 that the term "affect" would be more appropriate in the title. In particular, because overexpression of Ubc8 does not 'promote', but rather compromise protein import.

I would also like to second the following comment made by reviewer #3 regarding Figure 4A:

"Figure 4A: Western blots need to be quantified and normalized for loading to be convincing."

The figure displays a radiography of an import experiment. Given the current title "The metabolite-controlled ubiquitin conjugase Ubc8 promotes mitochondrial protein import", it is a central experiment to show that import is affected in the absence of Ubc8. Quantification of such an experiment performed in triplicates appears to be a very reasonable request.

Figures 1F and 1G don't appear to be in the correct order.

Reviewer #3 (Comments to the Authors (Required)):

The authors addressed by concerns.

September 30, 2022

RE: Life Science Alliance Manuscript #LSA-2022-01526-TRR

Dr. Johannes M Herrmann
University of Kaiserslautern
Cell Biology
Erwin-Schroedinger-Strasse 13
Kaiserslautern D-67663
Germany

Dear Dr. Herrmann,

Thank you for submitting your Research Article entitled "The metabolite-controlled ubiquitin conjugase Ubc8 promotes mitochondrial protein import". It is a pleasure to let you know that your manuscript is now accepted for publication in Life Science Alliance. Congratulations on this interesting work.

DISTRIBUTION OF MATERIALS:

Again, congratulations on a very nice paper. I hope you found the review process to be constructive and are pleased with how the manuscript was handled editorially. We look forward to future exciting submissions from your lab.

Sincerely,
